# Coherent backscatter enhancement in bistatic Ku-/X-band radar observations of dry snow

Marcel Stefko[1,*], Silvan Leinss[1,2,3,*], Othmar Frey[1,3], and Irena Hajnsek[1,4]

[1]ETH Zurich, Institute of Environmental Engineering, 8093 Zurich, Switzerland
[2]Université Savoie Mont Blanc, LISTIC, 74000 Annecy, France
[3]Gamma Remote Sensing, 3073 Gümligen, Switzerland
[4]German Aerospace Center, Microwaves and Radar Institute, 82234 Wessling, Germany
[*]These authors contributed equally to this work.

**Correspondence:** Marcel Stefko (stefko@ifu.baug.ethz.ch) and Silvan Leinss (leinss@gamma-rs.ch)

**Abstract.** The coherent backscatter opposition effect (CBOE) enhances the backscatter intensity of electromagnetic waves by up to a factor of two in a very narrow cone around the direct return direction when multiple scattering occurs in a weakly absorbing, disordered medium. So far, this effect has not been investigated in terrestrial snow in the microwave spectrum. It has also received little attention in scattering models. We present the first characterization of the CBOE in dry snow using ground-based and space-borne bistatic radar systems. For a seasonal snow pack in Ku-band (17.2 GHz), we found backscatter enhancement of 50–60% (+1.8–2.0 dB) at zero bistatic angle and a peak half-width-at-half-maximum (HWHM) of $0.25°$. In X-band (9.65 GHz), we found backscatter enhancement of at least 35% (+1.3 dB) and an estimated HWHM of $0.12°$ in the accumulation areas of glaciers in the Jungfrau-Aletsch region, Switzerland. Sampling of the peak shape at different bistatic angles allows estimating the scattering and absorption mean free paths, $\Lambda_T$ and $\Lambda_A$. In the VV polarization, we obtained $\Lambda_T = 0.4 \pm 0.1$ m and $\Lambda_A = 19 \pm 12$ m at Ku-band, and $\Lambda_T = 2.1 \pm 0.4$ m, $\Lambda_A = 21.8 \pm 2.7$ m at X-band, assuming an optically thick medium. The HH polarization yielded similar results. The observed backscatter enhancement is thus significant enough to require consideration in backscatter models describing monostatic and bistatic radar experiments. Enhanced backscattering beyond the Earth, on the surface of solar system bodies, has been interpreted as being caused by the presence of water ice. In agreement with this interpretation, our results confirm the presence of the CBOE at X- and Ku-band frequencies in terrestrial snow.

## 1  Introduction

The scattering of electromagnetic waves in any type of medium can be used to characterize some of its structural properties. In radar remote sensing, the scattering characteristics of snow have been intensely studied to derive properties of the snowpack. However, an important effect, the coherent backscatter opposition effect (CBOE), can enhance the radar backscatter return by up to a factor of two. This effect has rarely been considered in descriptions of the backscatter return from snow in monostatic radar experiments (where the transmitter and the receiver are co-located), because even though the CBOE is present, its mag-

nitude cannot be quantified without a bistatic reference measurement (where the transmitter and the receiver are at separate locations). To fully characterize the CBOE, bistatic radar experiments need to be performed.

## 1.1 Opposition effects in random media

An opposition effect (also referred to as "opposition peak", "opposition surge", "enhanced backscattering", "hot spot", and similar) is any phenomenon in which electromagnetic (EM) radiation scattered from a particular medium exhibits an increase in intensity in the direct return direction and its vicinity. Opposition effects occur at a variety of wavelengths and scattering media, and are caused by a variety of underlying physical phenomena (Hapke, 2012).

The *coherent* backscatter opposition effect (CBOE), also referred to as coherent backscatter enhancement or weak localiza-
tion of electromagnetic radiation, occurs when coherent EM radiation is scattered two or more times within a weakly absorbing, disordered medium. In the direct return direction, where wave vectors of the incident and scattered wave are parallel, the EM waves, travelling through the medium along each possible scattering path, interfere constructively with their time-reversed counterparts (Kuga and Ishimaru, 1984; Tsang and Ishimaru, 1984; Akkermans et al., 1988; Aegerter and Maret, 2009; Hapke, 2012). This constructive interference enhances the backscatter intensity within a very narrow cone of about 0.01–1 degree width
by up to a factor of two, whereas transmission is reduced. In all other directions, scattered waves sum up incoherently and form the incoherent background scatter signal. The angular half-width at half-maximum (HWHM) of the CBOE peak is proportional to the ratio of the free space wavelength $\lambda$ and the scattering mean free path $\Lambda_T$ (Hapke, 2012, Eq. 9.42). The peak tip can be very sharp when high orders of scattering contribute. The peak becomes rounder, wider and less intense when absorption and finite sample thickness limit the contribution of multiple scattering (Akkermans et al. (1988, Fig. 7), Van Der Mark et al.
(1988, Fig. 20)).

The CBOE can occur together with the shadow hiding opposition effect (SHOE). However, the SHOE requires particles to be large enough to cast sharp shadows within a porous medium (fine dust, vegetation canopy). Particles can then hide their own shadow in the direct return direction (Hapke, 1986; Hapke et al., 1996). In contrast to the CBOE, the SHOE is caused by single scattering while multiple scattering weakens the SHOE; absorption is not critical. The HWHM of the SHOE peak is given by
the ratio of particle radius and particle-to-shadow distance or extinction length in the medium (Hapke, 2012, Eq. 9.24). For the surface of the Moon, acting as a prototype for virtually all solar system objects with exposed surfaces, both the CBOE and the SHOE contribute with similar parts to the scattered light in the visible spectrum in the direct return direction (Hapke et al., 1998).

## 1.2 Observations of the CBOE

Most quantitative measurements of the CBOE that characterize the whole angular width of the peak have been carried out at visible-light wavelengths through laboratory experiments which are easier to realize than radio-frequency field- and planetary experiments (Montgomery and Kohl, 1980; Kuga and Ishimaru, 1984; Wolf and Maret, 1985; Van Albada and Lagendijk, 1985; Hapke, 1986; Akkermans et al., 1986; Wolf et al., 1988; Van Albada et al., 1988; Van Albada et al., 1990; Hapke et al., 1996; Mishchenko et al., 2000). In the context of the Earth's cryosphere, Kaasalainen et al. (2006) investigated and confirmed the

presence of 10–60% backscatter enhancement in snow at optical wavelengths (632.8 nm) and interpreted the observed narrow angular peak width of 0.1–1 degree at HWHM as dominated by the CBOE.

In the radio-frequency spectrum, the CBOE is mostly discussed in connection with snow and ice deposits where microwave absorption is weak (Warren and Brandt, 2008; Mätzler and Wegmüller, 1987a, b). In planetary science, the CBOE was proposed as an explanation for the unusually high radar cross-sections of surfaces of various solar system bodies (Muhleman et al., 1991; Black et al., 2001; Hapke et al., 1993). The CBOE was also discussed as a potential cause of the unusual radar echoes from the Greenland ice sheet (Rignot et al., 1993), although internal reflections were proposed as an alternative explanation (Rignot, 1995). In both of these contexts, additional measurements at small but nonzero bistatic angles were desired (but not feasible), as they would have provided a way to more easily and robustly characterize the effect (Hapke, 1990; Rignot, 1995). Bistatic radar measurements of surfaces of solar system bodies are possible by using an orbiting spacecraft in combination with the deep space network receivers on Earth (Simpson, 1993; Palmer et al., 2017). However, such experiments require a very specific geometric alignment of the spacecraft's orbit with respect to the Earth and are thus not common. Nevertheless, several experiments were carried out with the Moon as the target (Yushkova et al., 2018): the Clementine bistatic radar experiment observed an opposition peak in certain areas of the lunar surface. This peak was suggested to be attributable to the CBOE, implying the existence of ice deposits on the surface (Nozette et al., 1996), though other work called the interpretation of the Clementine data into question (Simpson and Tyler, 1999). More recently, the Mini-RF instrument of the Lunar Reconnaissance Orbiter, in concert with the Arecibo Observatory's radio telescope acting as transmitter, detected the opposition surge in certain areas of the lunar surface, again attributed to the presence of near-surface deposits of water ice (Patterson et al., 2017).

In many of these experiments, observation of a backscatter enhancement peak at radio-frequencies was interpreted as the CBOE. This interpretation was then used to infer the possible existence of water ice (presumably with a porous or disordered structure so as to elicit the effect) on the surface of the corresponding solar system bodies. Other works considered the CBOE in microwave scattering models of terrestrial snow (Tan et al., 2015) but could not analyze the peak shape of the CBOE. In this work we demonstrate that the existence of snow on the Earth can indeed cause a CBOE. We present a sampling of the peak shape at Ku- and X-band radio wavelengths with ground-based and space-borne imaging radars.

## 2  Methods

To characterize the angular peak of backscatter enhancement effects in the radio-frequency spectrum, we used two bistatic radar systems, the ground-based system KAPRI and the space-borne satellite formation TanDEM-X. For both systems, the transmitter and receiver are placed on independent platforms and thus the bistatic angle can be varied. The bistatic angle $\beta$ is defined as the angle between the transmitter, the observed location, and the bistatic receiver. In the exact direct return direction, the bistatic angle is zero and the scattering alignment is called the monostatic configuration.

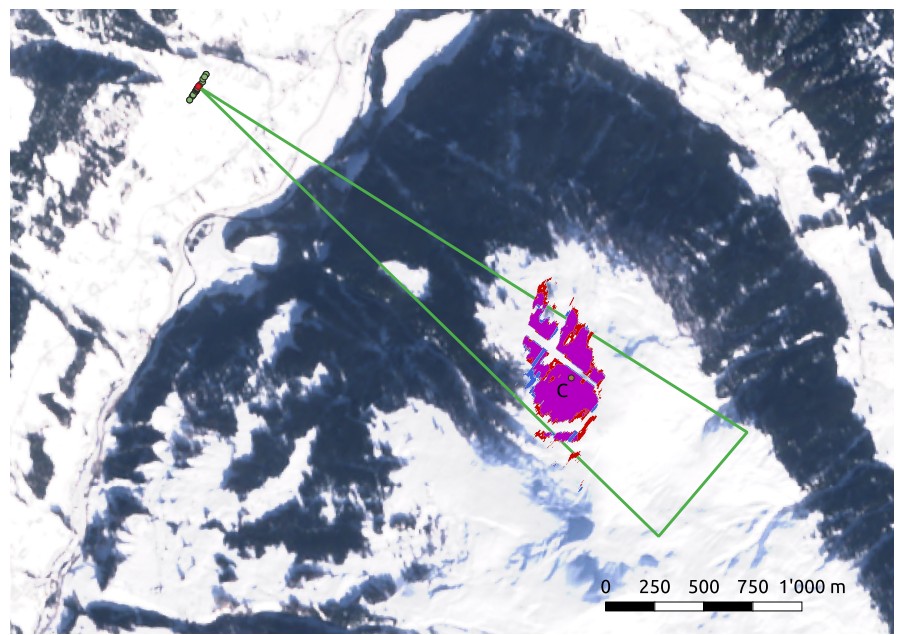

**Figure 1.** Map of the ground-based bistatic radar experiment. The position of the fixed transmitter (red dot) and the different receiver positions (green dots) are marked in upper left corner and form the bistatic baseline $b$. The green triangle marks the $-3\,\mathrm{dB}$ antenna beamwidth of $12°$ of the receiver device. The antennas are oriented towards the snow covered north-west face of the mountain Rinerhorn, Switzerland. The region of interest (ROI) for the winter and summer seasons is shown in blue and red respectively. Their overlap is shown in purple. C is a reference point for the orientation of the bistatic receiver (see Fig. 4). The hollow line slicing through the ROI masks out metallic beams from a ski-lift on the slope. Satellite imagery data: Sentinel-2 on 20 February 2021. Modified Copernicus Sentinel data 2021/Sentinel Hub.

## 2.1 Ground-based observations - KAPRI

KAPRI is a polarimetric radar system based on the GPRI, developed by Gamma Remote Sensing (Werner et al., 2012; Baffelli et al., 2017). It is a ground-based Ku-band FMCW real-aperture radar system, capable of performing fully-polarimetric, bistatic measurements. In the bistatic configuration comprised of two synchronized radar instruments with different antenna configurations, the bistatic system offers coverage of areas hundreds of meters wide at a range of several kilometers. The instruments operate at a central frequency of $17.2\,\mathrm{GHz}$ ($\lambda = 1.74\,\mathrm{cm}$), with $200\,\mathrm{MHz}$ bandwidth. The bistatic configuration and the processing pipeline to generate bistatic single look complex (SLC) data are detailed in (Stefko et al., 2022). A description of the antenna configuration while using a cable synchronization setup can be found in (Stefko et al., 2021, Fig. 2).

### 2.1.1 Observation site: Rinerhorn, Davos

For the ground-based experiment (map shown in Fig. 1), the observed region of interest (ROI) was located on the north-western face of Rinerhorn peak near Davos, Switzerland. Both devices were located on the valley side opposite the peak, at 46.763 N, 9.788 E (Fig. 1). The radar location features a straight and relatively flat segment of road approximately $200\,\mathrm{m}$

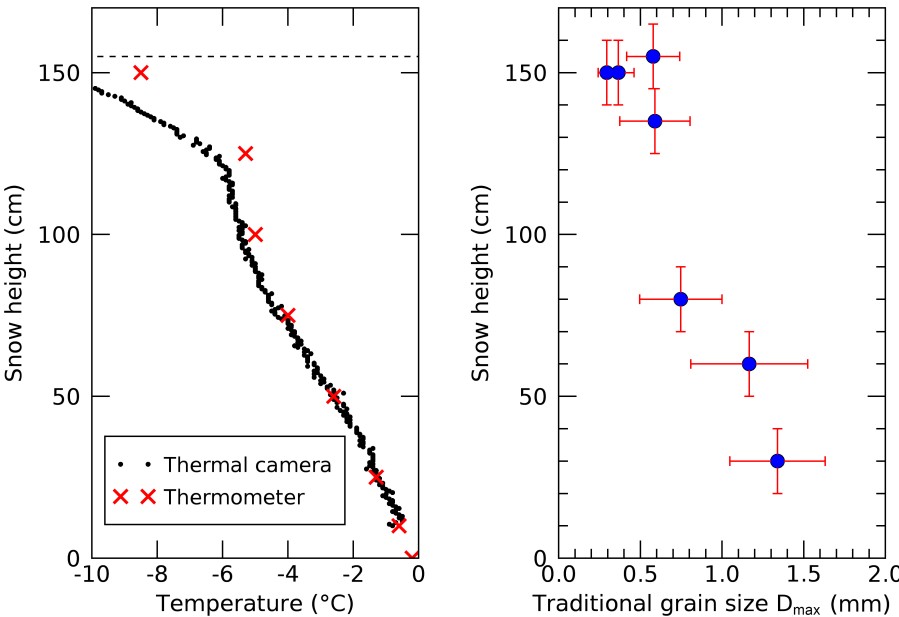

**Figure 2.** Snow temperature and grain size in the study area close to point C in Fig. 1 on 18 February 2021. Snow height is 1.55 m.

long with unobstructed view of Rinerhorn. The devices were placed at approximately 1620 m altitude, while the ROI altitude spans from 2050 m to 2270 m. With this upward-looking observation geometry the vast majority of the ROI area is observed under a shallow local incidence angle larger than 70°. Problems with multipath interference arising from the upward-looking observation geometry while employing a fan-beam radar system (Lucas et al., 2017) are avoided by placing the instruments on the opposing side of the valley.

We performed two experiments: in summer (05 August 2020), the ROI was covered by low grass. In winter (18 February 2021), the area was completely covered by approximately 1.5 m of seasonal snow. Each measurement began at approximately 08:00 local time, and the total duration of the observations was 3.5 hours in summer, and 5.5 hours in winter. In winter, a snow pit revealed snow temperatures of $-10\,°C$ at the snow surface and $-0.2\,°C$ at the bottom of the snowpack (Fig. 2, left). The traditional snow grain size, measured as the mean maximum extent of snow crystals (Fierz et al., 2009), was $D_{max} = 0.3\,mm$ at the surface and 1.5 mm at the base (Fig. 2, right).

To select the ROI, a mask fulfilling the following three conditions was applied for each season: 1) Include only terrain higher than the tree line at 2050 m altitude. 2) Exclude areas containing man-made structures (metallic support beams, metal ropes, buildings, corner reflectors). 3) Exclude areas affected by radar shadow and exclude areas outside of the main beam of the secondary antennas – these areas were detected by applying a threshold on the magnitude of the single-pass interferometric coherence $\gamma$ of the secondary receiver in the VV channel. In every acquisition in the summer dataset, pixels with $\gamma < 0.85$ were masked out, in the winter dataset pixels with $\gamma < 0.80$ were masked out. A sliding window of $5 \times 3$ (range $\times$ azimuth) pixels was used for coherence estimation. The winter threshold is lower due to lower overall coherence as opposed to summer.

The masks defining the ROI for each season are shown in Fig. 1. They cover practically the same region of the hillside. The acquired calibrated SLC datasets were spatially multi-looked using a $5 \times 3$ window to obtain the intensity images, and analyzed in the radar polar geometry (range $\times$ azimuth angle). The intensity value $\hat{I}(\beta)$ (the hat symbol $\hat{}$ indicates it's a measured quantity) was computed for every acquisition by averaging the measured intensities of all pixels within the ROI. We analyzed only acquisitions with VV and HH polarization as the cross-polarized signal was too close to the noise floor to provide reliable data.

### 2.1.2 Device configuration and measurement procedure

The primary (monostatic) transmitter-receiver remained stationary during the experiment (Fig. 3, top) and performed azimuthal sweep acquisitions of the observed area at a range of approximately 2.5 km. The secondary device (bistatic receiver) was moved step-wise to sample bistatic angles between $0.04°$ and $1.92°$. In winter, the secondary device was mounted on a large sledge (Hornschlitten, Fig. 3, middle). In summer, a wheeled cart was used as a movable radar platform (Fig. 3, bottom). The trajectory of the secondary device is visualized in Fig. 4.

The bistatic angle $\beta$ was calculated for each position of the secondary receiver S as

$$\beta = \arctan \frac{b}{d_{\mathrm{PC}}}. \tag{1}$$

The length of the bistatic baseline $b$ is given by the length of the vector between the primary and secondary radar's positions, after projecting it into the plane orthogonal to the line of sight. The line of sight is the vector between the primary radar P and the reference point C in the ROI (see Fig. 4). Its length is $d_{\mathrm{PC}} = 2500\,\mathrm{m}$.

To ensure optimal overlap of the antenna patterns in the ROI, the secondary device was leveled and oriented manually in each position. The pitch and roll angles of the mobile platform with respect to the true vertical direction were measured with a digital spirit level at each measurement point, and did not exceed $2°$ in either direction. In azimuth, the device was oriented with a compass and optical viewfinder using a reference point in the center of the ROI (Point C in Fig. 1). The estimated precision is $1°$.

The transmit antennas on the primary KAPRI device have a physical horizontal length of 2 m (Fig. 3, top), and thus the bistatic angle at $2.5\,\mathrm{km}$ range differs by $\sim 0.05°$ between the two edges of the transmit antenna. This imposes a practical limit on the resolution of the sampling of the intensity curve: any variation of intensity within bistatic angles of less than $0.05°$ will be smeared out by the non-zero size of the transmit antennas.

### 2.1.3 KAPRI: radiometric precision

Three main factors were identified which can affect the radiometric precision of the measurements: temporal drift of the scattering properties in the ROI, the radiometric stability of the bistatic KAPRI system, and the pointing precision of the secondary receiver's antennas.

The trajectory of the bistatic receiver (Fig. 4) was designed to repeatedly increase and decrease the absolute value of the bistatic baseline. This allows detection of any temporal drift of the scattering intensity over the course of the measurement (i.e.

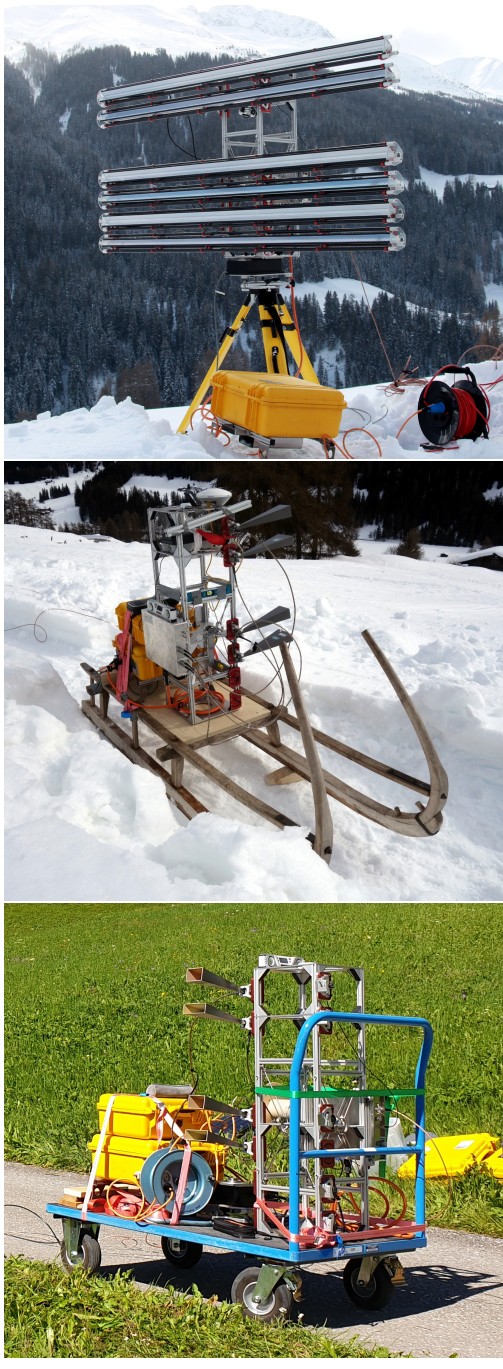

**Figure 3.** Top: primary KAPRI radar tower (monostatic device) during the ground-based experiment equipped with narrow-beam traveling-wave antennas. Center: secondary KAPRI radar tower (bistatic receiver) equipped with horn antennas and deployed on a "Hornschlitten" sledge in winter. Bottom: secondary KAPRI tower deployed on a wheeled cart in summer.

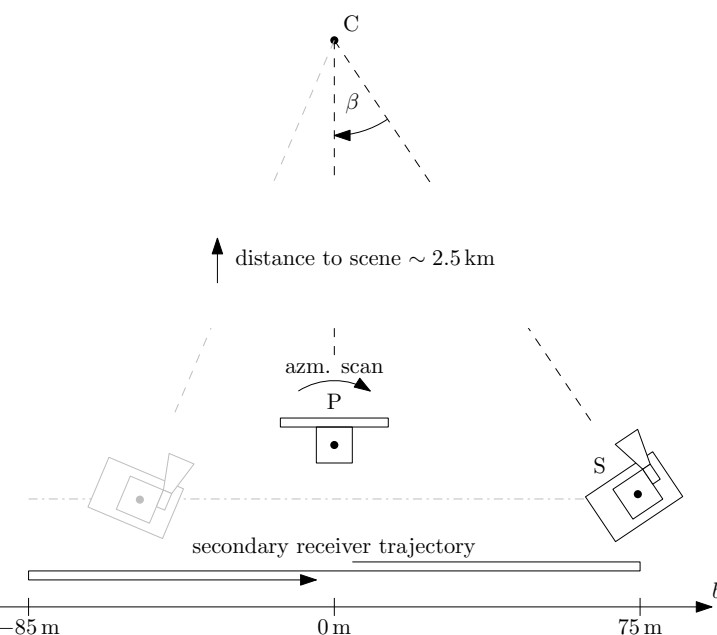

**Figure 4.** Diagram of the ground-based measurement procedure. The stationary, primary device P performs repeated azimuthal scans of the ROI around point C. To sample the scattering response of the ROI under a variety of bistatic angles $\beta$, the mobile secondary device S is repositioned in-between acquisitions. For both winter and summer experiments, S was placed at bistatic baselines $b$ varying between $-85$ and $+75\,\mathrm{m}$ relative to P.

on the order of minutes to hours). Drifts would be detected by the different shape of the left and right wing of the intensity curve $\hat{I}(\beta)$.

The radiometric stability of KAPRI can be assessed by investigating the monostatic scattering intensity observed by the monostatic device from a reference target (a corner reflector). The maximal detected variation was observed in the HH channel in the winter season, with standard deviation of $16\,\%$ relative to the mean value.

For each individual measurement the beam pointing direction of the secondary receiver differed by less than $1°$ in azimuth from the ideal central pointing direction towards point C. Due to the antenna pattern of the secondary receiver (Stefko et al., 2022, Fig. 7), an azimuthal misalignment of $1°$ can reduce the signal intensity by not more than $\sim 1\,\mathrm{dB}$ (25%) at the edge of the "ideal" antenna pattern footprint covering the ROI. However, when considering the total received backscatter from the ROI, this reduction is partially compensated, since the observed backscatter intensity from the other edge of the ROI would necessarily increase.

Due to the limited radiometric stability and the beam pointing uncertainty, the observed backscatter intensity can thus be expected to vary stochastically with estimated standard deviation of approximately $20\,\%$, affecting each individual measurement by a significant amount. These effects are difficult to compensate for, since there were no reference targets in the scene with a sufficiently high and stable bistatic radar cross-section. For this reason, no a-posteriori radiometric calibration was applied

to the data. However, the two effects are stochastic in nature, and uncorrelated between individual receiver positions, and thus with a sufficiently high number of acquisitions, the enhancement peak should still be detectable, albeit with lower radiometric precision.

## 2.2 Satellite observations - TanDEM-X

The TanDEM-X satellite formation is the first space-borne bistatic radar system with an adjustable bistatic baseline. The formation consists of two free flying synthetic aperture radar (SAR) satellites, TerraSAR-X and TanDEM-X, orbiting the Earth in about 514 km height in a helix-like formation (Krieger et al., 2007). The two radar instruments operate at X-band at a central frequency of 9.65 GHz ($\lambda = 3.11$ cm). Depending on the acquisition mode, both satellites can act as either transmitter or receiver or both. In the bistatic mode, the transmit-receive satellite operates in a monostatic observation geometry, and the receive-only satellite operates in a bistatic observation geometry.

Since the launch of TanDEM-X in June 2010, the distance between the two satellites was varied by several kilometers. The largest (and smallest) distances were obtained during the TanDEM-X science phase between Oct 2014 and February 2016 (Hajnsek et al., 2014). To find an area best suited for observation of the CBOE in X-band, we searched the entire TanDEM-X archive for areas that are covered by deep snow and where long acquisition time series with large bistatic angles are available. Unfortunately, near the poles, bistatic angles are relatively small, making a sufficient sampling of the CBOE peak difficult. At the equator, the largest bistatic angles of up to $0.35°$ are available but snow is naturally rare. As a best compromise, we selected the Jungfrau-Aletsch region in Switzerland but also analyzed the Teram-Shehr/Rimo glacier in the Karakorum (supplementary material) where a considerably lower number of acquisitions was available.

### 2.2.1 Jungfrau-Aletsch region

The Jungfrau-Aletsch region was selected as a TanDEM-X super test site with the aim to acquire as many acquisitions as possible, and to explore the scientific value of the bistatic radar mission. 118 bistatic acquisitions at two polarizations (VV, HH) were acquired between 2011 and 2019, most of them during winter (Fig. 5). At VH polarization no acquisitions at sufficiently large $\beta$ were available. For 104 acquisitions TerraSAR-X acted as transmitter, for 14 acquisitions TanDEM-X acted as transmitter. We removed the 14 TanDEM-X acquisitions because they showed slightly different antenna patterns that could not be compensated through the calibration, especially at HH polarization, because of a too small number of acquisitions. For the remaining 104 acquisitions, bistatic baselines between 65 and 2100 m are available, corresponding to $\beta = 0.005-0.21°$. The incidence angle at the scene center is $\theta = 32°$ (orbit 154, descending). Time-averaged backscatter images of the study area are provided in the supplementary figures S2 and S3. Interferometric and polarimetric properties of the dataset were analyzed by Leinss and Bernhard (2021).

The Jungfrau-Aletsch region is highly glaciated with multiple peaks reaching above 4000 m. Cold firn, several tens of meters deep, is likely present throughout the year: depending on exposition, the transition to temperate firn is at 3400–4000 m while the upper 15 m of firn experience seasonal temperature cycles and can freeze in winter (Suter et al., 2001; Suter and Hoelzle, 2002; Jun et al., 2002). At the end of March 2021, snow temperatures of $-11 \pm 3°$C in the upper two meters, and $-4 \pm 2°$C

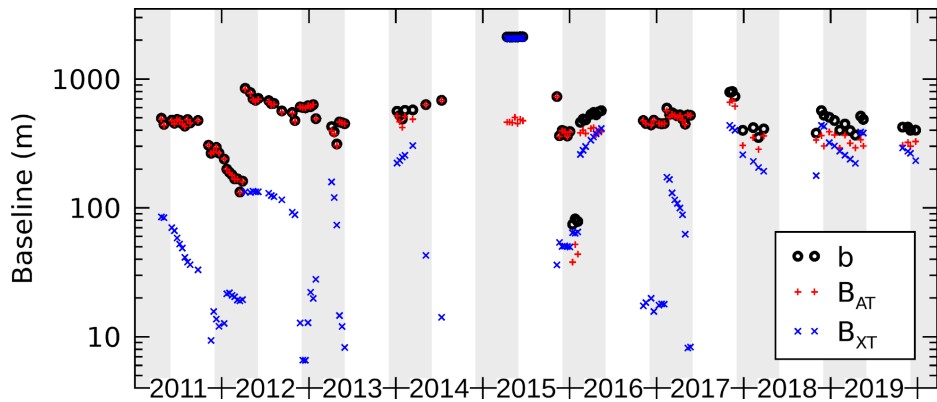

**Figure 5.** Time series of the bistatic baselines $b$ according to Eq. (2) together with along- and across-track baselines $B_{AT}$ and $B_{XT}$ of the TanDEM-X satellite acquisitions of the Jungfrau-Aletsch region. Along-track baselines are adjusted by 30 m due to the satellite motion (see Sect. 2.2.2). Gray shading indicates the period from 1 December until 31 May for which we assume that the firn, present in the accumulation area > 3500 m, is completely frozen.

at -8 m, were measured by Bannwart (2021) at two sites, one at 3380 m altitude (46.5525°N, 8.0286°E) and at 3350 m altitude (46.5483°N, 8.0323°E). At the beginning of March 2022, we measured snow temperatures of $-12 \pm 3$ °C in the upper two meters and $-4 \pm 1$ °C at -5 m at 3640 m altitude (46.5515°N, 8.0062°E). Both Bannwart's firn cores as well as our snow pit measurements indicate the presence of a few cm thick ice layer resulting from melt and refreeze during previous summers below the several meter thick seasonal snow cover.

The region contains Great Aletsch Glacier (46.50°N, 8.03°E), the largest glacier in the European Alps. Its equilibrium line altitude, above which accumulation dominates, is at ˜3000m (Zemp et al., 2007). In the ablation area below, seasonal snow is present. Ice free areas are dominated by rock and scree. Below 2500 m vegetation dominates with a tree line of 2000 m. For analysis of specific land cover types, we selected the following three regions of interest (ROI) in the Jungfrau-Aletsch region (shown also in the supplementary figure S8):

1. The accumulation area of glaciers with altitudes above 3500 m. These areas are at or above the temperate-to-cold firn transition and we assume that firn conditions did not change too drastically from winter to winter. To ensure refreezing of firn after summer, and to avoid snow melt in spring, we restricted the model parameter estimation on data acquired between 01 December and 31 May (gray shading in Fig. 5). The dry, deep firn acts as a thick medium with multiple scattering in the volume but low absorbing.

2. The ablation area of Great Aletsch Glacier with altitudes below 2700 m. Field measurements indicate a seasonal snow cover of 0–3 m on the glacier tongue during winter (Leinss and Bernhard, 2021). The seasonal snow acts as a thin layer of volume scatterers with low absorption if the snow is dry ($T < 0$°C).

3. Forested areas with at least 7 m height, mainly conifer forest located in the Rhone valley and the Grindelwald region. The Forest acts as a medium where both volume scattering and absorption are significant.

### 2.2.2 TanDEM-X: bistatic angle

For TanDEM-X, the bistatic angle $\beta = b/R$ was determined from the average slant-range distance $R$ to the scene center and from the bistatic baseline $b$, derived from the orbit coordinates. To compute $b$, the distance between the two satellites was decomposed into the along-track baseline $B_{\mathrm{AT}}$, the across-track baseline $B_{\mathrm{XT}}$ and the parallel, or line-of-sight baseline baseline $B_{\mathrm{LOS}}$. The bistatic baseline $b$, perpendicular to the line-of-sight direction, is given by

$$b = \sqrt{B_{\mathrm{AT}}^2 + B_{\mathrm{XT}}^2}. \tag{2}$$

Figure 5 shows time series of $b, B_{\mathrm{AT}}$, and $B_{\mathrm{XT}}$. Because of the bistatic acquisition geometry, where the phase center of the bistatic receiver is located in the midpoint between the transmitter and the receiver (Duque et al., 2012), the across and along-track baselines used in Eq. (2) are a factor of two larger than the effective interferometric across- and along-track baselines given in the acquisition's meta-information (Leinss and Bernhard, 2021, cf. Fig. 2).

Even though we refer in the following to the monostatic acquisition, we note, that the orbital velocity of $v = 7.6\,\mathrm{km\,s}^{-1}$ results in a small, velocity induced, bistatic angle of $\beta_v = 0.003°$ for the monostatic receiver, because the satellite moves 30 m between transmission and reception of a radar pulse. The high orbital velocity also decreases (increases) the along-track baseline $B_{\mathrm{AT}}$ by 30 m when the bistatic receiver follows (is ahead of) the transmitter. We considered this in the analysis but found the effect negligible.

### 2.2.3 TanDEM-X: radiometric calibration and computation of backscatter ratios

Resolving the peak shape of the CBOE with a maximum expected peak height of 3 dB requires a precise radiometric calibration of the bistatic dataset. To avoid any terrain or incidence angle dependent calibration, we analyzed the ratio between the backscatter intensity $\hat{I}_{\mathrm{bist}}$ observed by the bistatic receiver and the intensity $\hat{I}_{\mathrm{mono}}$ observed by the monostatic transmitter-receiver:

$$\hat{I}_{\mathrm{r},0} = \hat{I}_{\mathrm{bist}}/\hat{I}_{\mathrm{mono}}. \tag{3}$$

The index r,0 indicates taking the ratio relative to $I(\beta = 0)$. Averaging ratios like $\hat{I}_{\mathrm{r},0}$ would result in a biased estimate. To estimate unbiased spatially or temporally averaged ratios we first applied the averages on $\hat{I}_{\mathrm{bist}}$ and $\hat{I}_{\mathrm{mono}}$ and then computed the ratio $\hat{I}_{\mathrm{r},0}$. We use $\hat{I}$ to refer to the radar brightness commonly denoted by $\beta_0$ (Raney et al., 1994) to avoid confusion with the bistatic angle $\beta$.

For each polarization channel, we coregistered time series of the interferometric TanDEM-X CoSSC (Coregistered Single look Slant range Complex) acquisition pairs (Fritz et al., 2012; Duque et al., 2012). To obtain the intensities $\hat{I}_{\mathrm{mono}}$ and $\hat{I}_{\mathrm{bist}}^{\mathrm{uncal.}}$, we detected the temporally coregistered CoSSCs, applied $10 \times 10$ pixels multilooking, and downsampled the data by a factor of 10.

Unlike the monostatic products, the bistatic TanDEM-X products are not radiometrically calibrated (Fritz et al., 2012, Sect. 4.3). The intensity ratio $\hat{I}_{\mathrm{r,0}}^{\mathrm{uncal.}} = \hat{I}_{\mathrm{bist}}^{\mathrm{uncal.}}/\hat{I}_{\mathrm{mono}}$ showed, therefore, differences of 10–30% between the bistatic and the monostatic receiver. While at VV polarization $\hat{I}_{\mathrm{r,0,VV}}^{\mathrm{uncal.}}$ showed spatially relatively constant values at small bistatic angles, $\hat{I}_{\mathrm{r,0,HH}}^{\mathrm{uncal.}}$ showed terrain-independent trends of a few percent, presumably due to different antenna patterns (supplementary Figs. S4 and S5). To compensate for these patterns, we calibrated the intensity $\hat{I}_{\mathrm{bist}}^{\mathrm{uncal.}}$ at each polarization with the ratio of the pixel-wise temporal mean $\langle \cdot \rangle^{\mathrm{temp.}}$ of 17 scenes with $\beta < 0.033°$. This threshold for $\beta$ was chosen small enough to avoid any significant differences of backscatter enhancement between the monostatic and bistatic receiver. The bistatic intensity after antenna calibration is

$$\hat{I}_{\mathrm{bist}}^{\mathrm{ant.cal.}} = \hat{I}_{\mathrm{bist}}^{\mathrm{uncal.}} \frac{\langle \hat{I}_{\mathrm{mono}} \rangle_{\beta < 0.033°}^{\mathrm{temp.}}}{\langle \hat{I}_{\mathrm{bist}}^{\mathrm{uncal.}} \rangle_{\beta < 0.033°}^{\mathrm{temp.}}} \tag{4}$$

To obtain the calibrated intensity $\hat{I}_{\mathrm{bist}}$, we compensated in each acquisition pair for the remaining spatially constant offset between the monostatic and bistatic data. For this we multiplied $\hat{I}_{\mathrm{bist}}^{\mathrm{ant.cal.}}$ with the ratio of the monostatic and bistatic radar brightness, spatially averaged, as indicated by $\langle \cdot \rangle_{\mathrm{cal.area}}^{\mathrm{spat.}}$, over a pre-defined calibration area:

$$\hat{I}_{\mathrm{bist}} = \hat{I}_{\mathrm{bist}}^{\mathrm{ant.cal.}} \frac{\langle \hat{I}_{\mathrm{mono}} \rangle_{\mathrm{cal.area}}^{\mathrm{spat.}}}{\langle \hat{I}_{\mathrm{bist}}^{\mathrm{ant.cal.}} \rangle_{\mathrm{cal.area}}^{\mathrm{spat.}}} . \tag{5}$$

For calibration of the Jungfrau-Aletsch dataset we used areas that showed a temporally stable and baseline-independent backscatter ratio $\hat{I}_{\mathrm{r,0}}$. These areas were defined using two iterations. In a first iteration, we masked out very dark areas, possibly affected by noise, such as shadow ($\hat{I}_{\mathrm{mono}} < -14\,\mathrm{dB}$) and also very bright areas such as layover and strong local scatterers ($\hat{I}_{\mathrm{mono}} > +1\,\mathrm{dB}$) through thresholding the temporal mean of the backscatter intensity. We also masked out the ROIs later analyzed, by masking elevations above 3000 m where multi-year firn occurs, regions covered by forest, as well as the ablation area of Great Aletsch Glacier. After using the remaining pixels for calibration, in the second iteration we masked out additionally areas, showing possible artifacts, where $\hat{I}_{\mathrm{r,0}}$, computed pixel-wise using the temporal means of $\hat{I}_{\mathrm{mono}}$ and $\hat{I}_{\mathrm{bist}}$ from 43 acquisitions with bistatic angles smaller than $0.04°$ (cf. Eq. (4)), deviated more than 5% from unity. Such deviations appeared in areas of low radar backscatter and areas not directly affected by layover but next to layover in the far-range direction. The deviations might partially originate from bright azimuth ambiguities. We also believe that double-reflections occurring within layover, with a reflection on each side of a north-south oriented valley and an additional propagation path between the two reflections, cause further radar echos appearing beyond the layover area. These artifacts appear stronger at HH than at VV (due to reflections close to the Brewster-angle) and are also stronger with wet snow due to more specular reflection compared to dry snow or summer with more diffuse reflections (the artifacts are well visible in supplementary figures when comparing S2 with S3 and S4 with S5). We also removed areas where the pre-calibrated backscatter ratio $\hat{I}_{\mathrm{r,0}}$ showed a temporal standard deviation larger than $0.08$ (supplementary figures S6 and S7). Finally, to avoid that the CBOE or possibly the SHOE affect the calibration, we masked out areas that showed more than 5% enhanced scattering in the direct return direction in the large-baseline acquisitions $B_\perp > 2\,\mathrm{km}$. In total, we masked out approximately 50% of pixels from the scene (supplementary Fig. S8) and used the remaining pixels, mainly grassland, rock and the ablation areas of glaciers for calibration in Eq. (5). In this data-driven calibration we assume that the regions selected for calibration show an equal backscatter intensity for the monostatic and bistatic receiver.

To determine the backscatter ratio for the ROIs, we used Eq. (3) with $\hat{I}_{\text{bist}}$ and $\hat{I}_{\text{mono}}$ averaged over the ROI. To differentiate between dry and wet snow for snow covered areas, we used the mean backscatter intensity $\hat{I}_{\text{mono}}$ as a proxy.

To display imagery of $I_{\text{r},0}$ with sufficient radiometric resolution, we applied additional $4 \times 4$ px multilooking to the downsampled backscatter imagery, corresponding to an effective multilooking operation of $41 \times 41$ pixels. This value was chosen to keep the standard deviation $\sigma = I/\sqrt{N}$ of the multilooked intensity $I$ sufficiently low (Oliver and Quegan, 2004). $N$ is the number of looks. Given that adjacent pixels are statistically not completely independent (the SLC data is oversampled by a factor of 1.3 in slant range and and 2.9 in azimuth, resulting in 3.73 pixels per look) we obtain a value of $N = 41^2/3.73 = 450$ looks which corresponds to a radiometric accuracy (standard deviation) of $0.2\,\text{dB}$ (5%) at an intensity of $I = -5\,\text{dB}$.

## 2.3  Backscatter model for the CBOE

Coherent backscatter enhancement was first explained through time-reverse propagation in double and multiple scattering paths between scatterers with a low volume fraction in free space using second order multiple-scattering theory and expansion in Feynman diagrams (Tsang and Ishimaru, 1984, 1985; Van Der Mark et al., 1988); Wolf et al. (1988) added particle-independent absorption through the background medium. For a review see (Akkermans et al., 1988) and the book from Hapke (2012). To our knowledge, no complete theory for CBOE in densely-packed media of particles small compared to the wavelength exists. Furthermore, in snow, scattering can occur at various length scales (i.e. ice grains, density fluctuations, inter-layer boundaries, and ice layers (Picard et al., 2018)) and no CBOE model for multi-layer structures is currently available. In order to describe the complex snow structure in the context of existing models, we consider snow as an effective scattering medium occupying a semi-infinite space with homogeneous scattering and absorption properties and follow the description from Hapke (2012) for interpretation and modeling of our results: in Chapter 9, Eqs. 9.40 and 9.44 (Hapke, 2012), as well in (Akkermans et al., 1986, 1988; Akkermans and Montambaux, 2004), the peak shape of the coherent backscatter enhancement is described for non-absorbing and absorbing media by the equation:

$$B_C(\beta) = \frac{1}{[1 + 1.42K]\,[1 + \xi(\beta)]^2}\left[1 + \frac{1 - e^{-1.42K\xi(\beta)}}{\xi(\beta)}\right] \tag{6}$$

where $B_C(\beta)$ is the magnitude of the coherent backscatter intensity enhancement relative to the incoherent background $I_0$ at small bistatic angles $\beta \approx \sin\beta$. For notational simplicity, and in accordance with (Hapke, 2012, Eq. 9.44) and (Wolf et al., 1988), we defined

$$\xi(\beta) = \sqrt{\left(\frac{2\pi\Lambda_T\beta}{\lambda}\right)^2 + \frac{3\Lambda_T}{\Lambda_A}}. \tag{7}$$

In this equation $\lambda$ is the free space wavelength, $\Lambda_T \propto S^{-1}$ is the transport mean free path which is proportional to the inverse of the scattering coefficient $S$ of the medium, and $\Lambda_A = A^{-1}$ is the absorption mean free path in the medium with absorption coefficient $A$. Assuming that the snow depth is much larger than $\Lambda_T$, i.e. that snow can be considered as an optically thick medium, the scattering and absorption coefficient that parameterize Eq. 7, can be linked to snow properties derived from density and the microstructure (Picard et al., 2018) as discussed in Sect. 4.4. The factor $K$ is a correction factor, described in (Hapke,

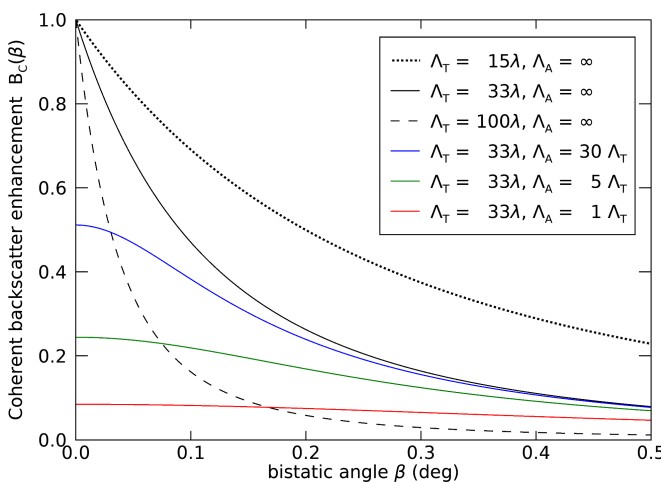

**Figure 6.** Modeled peak shape of the CBOE for different scattering mean free paths $\Lambda_T$ given in multiples of the wavelength $\lambda$ and for different absorption mean free paths $\Lambda_A$ (in multiples of $\Lambda_T$) in a medium with small scattering particles (porosity coefficient $K = 1$). For a non-absorbing medium ($\Lambda_A = \infty$) a very sharp peak can be observed. Already with a weak absorption $\Lambda_A = 30\Lambda_T$ (blue) the peak height is reduced to 50% and the peak becomes much rounder. For comparable scattering and absorption lengths the peak is not noticeable (red).

2012, p.164–167) as "porosity coeffcient". The factor $K$ increases the extinction coefficient $E = S + A$ in densely packed media where inter-particle effects of particles large relative to $\lambda$ occur. As ice grains are much smaller than the wavelength, we assume $K = 1$.

The incoherent background intensity $I_0$ is determined by the single- and multiply scattered background intensity from the medium for which no time-reverse counterparts exist (i.e. no coherent enhancement), so that

$$I(\beta) = I_0[1 + B_C(\beta)] \tag{8}$$

describes the total backscatter intensity $I(\beta)$ in the proximity of several degrees from the direct backscatter direction.

The peak shape, as drawn in Figure 6, is determined by the ratio of scattering mean free path $\Lambda_T$ to the wavelength $\lambda$, as
already indicated by Tsang and Ishimaru (1984), and by the probability distribution of scattering path lengths in the medium. A (monostatic) scattering path begins at the first scattering event in the medium, travels along multiple scatter events with mean distance $\Lambda_T$, and ends when the radiation is scattered back out of the medium in the direct return direction (Hapke, 2012, Chapter 9.3). In the monostatic configuration, radiation traveling along such a path interferes constructively with radiation propagating along the time-reversed counterpart, thus causing the backscatter intensity enhancement. Long scattering paths,
consisting of multiple scattering events, have a longer distance between the path's start and end point and cause a narrow peak, while short scattering paths cause a broad peak. The final peak shape is determined by the sum of all occurring peak shapes of different widths (Tsang and Ishimaru, 1985), weighted according to their occurrence probabilities. The more absorption occurs, the shorter are the scattering paths that can contribute to the coherent peak, and the lower is the probability for the occurrence of higher order scattering, hence the peak becomes rounder and wider (Akkermans et al., 1988; Wolf et al., 1988,

Fig. 7). Long scattering paths can also be limited by a finite sample (snowpack) thickness, which also causes a rounding of the peak and an increase of its width (Van Der Mark et al., 1988; Van Albada et al., 1988, Fig. 20).

Figure 6 shows the shape of the CBOE peak for a range of values of $\Lambda_T, \Lambda_A$ given in multiples of $\lambda$. For non-absorbing media ($\Lambda_A = \infty$, black curves), longer scattering lengths $\Lambda_T$ cause a narrower peak with a HWHM of $0.36\lambda/(2\pi\Lambda_T)$ (Van Der Mark et al., 1988; van Albada et al., 1987). This peak width holds for sparsely packed media; for densely packed media of hard spheres, (Mishchenko, 1992a) suggests a significantly reduced HWHM.

With increasing absorption, the peak height decreases, its width increases and the peak becomes rounder. To characterize the peak height and width for absorbing media, we found, that Eq. (6), with $K = 1$ can be well approximated by

$$B_C(\beta) \approx \frac{1}{\left[1 + 1.3\xi(\beta)\right]^2} \tag{9}$$

where the factor 1.3 corrects deviation resulting from neglecting 1st and 2nd order terms of $\xi(\beta)$ in the numerator. Eq. (9) provides an analytical form to link the ratio $\Lambda_T/\Lambda_A$ to the peak height

$$B_C(0) = \frac{1}{\left(1 + 1.3\sqrt{3\frac{\Lambda_T}{\Lambda_A}}\right)^2}. \tag{10}$$

A slightly more complicated equation can be obtained for the peak width for finite $\Lambda_A$. Hence, when characterizing the full peak shape, or at least its height and width, the parameters $\Lambda_T$ and $\Lambda_A$ can be determined.

Most CBOE models are based on scalar waves which do not consider the vector character of electromagnetic waves, i.e. their polarization. However, experimental and theoretical works show that CBOE occurs predominantly for co-polarized transmitted and received waves (VV and HH) where the model matches well experimental observation. They also show that CBOE for cross-polarized (VH) observations is significantly weaker and decreases with increasing sample thickness (van Albada et al., 1987; Mishchenko, 1992b, c; Wolf and Maret, 1985).

### 2.3.1 Application to KAPRI data

With the two ground-based KAPRI instruments, the benefit of the flexible configuration allows us to sample the intensity peak up to relatively high value of bistatic angle $\beta$, and thus the flat region of the intensity curve ($I(\beta \to \infty) \to I_0$) should be observable. However, the very top of the peak is difficult to sample due to the non-negligible size of the primary device's antennas, as well as the possibility of the devices obstructing each other's view when placed very close together. Because of this, for analysis of KAPRI data, we use the intensity ratio $I_{r,\infty}(\beta)$ which is normalized to the incoherent background intensity $I(\infty)$, and can be expressed with aid of Eq. (8) as:

$$I_{r,\infty}(\beta) = \frac{I(\beta)}{I(\infty)} = \frac{I_0(1 + B_C(\beta))}{I_0(1 + B_C(\infty))} = 1 + B_C(\beta) \tag{11}$$

To calculate the intensity ratio of Eq. (11) from the actual observed mean ROI intensity $\hat{I}(\beta)$, we approximate $I(\infty)$ as the mean value of $\hat{I}(\beta)$ from all acquisitions within the corresponding dataset where $\beta > 1°$ (i.e. values well within the flat region

of the intensity curve):

$$\hat{I}_{r,\infty}(\beta) \approx \frac{\hat{I}(\beta)}{\langle \hat{I}(\beta) \rangle_{\beta > 1°}} \tag{12}$$

Because the model $B_C = B_C(\beta, \Lambda_T, \Lambda_A)$ depends on the transport mean free path $\Lambda_T$ and absorption length $\Lambda_A$ through Eqs. (6) and (7), we can use Eq. (11) to fit different values of $\Lambda_T, \Lambda_A$ to the observed intensity curve $\hat{I}_{r,\infty}(\beta)$ by nonlinear least square minimization. For the fitting procedure, we used the TRF (trust region reflective) optimization method implemented via the `curve_fit` function of the `scipy.optimize` library (Virtanen et al., 2020). The initial parameter values of the $(\Lambda_T, \Lambda_A)$ pair were set to $(1\,\mathrm{m}, 100\,\mathrm{m})$, and both parameters were restricted to the non-negative real number domain.

### 2.3.2 Application to TanDEM-X data

With TanDEM-X we measured the intensity ratio $\hat{I}_{r,0}(\beta)$ between the bistatic receiver $\hat{I}_{\mathrm{bist}} = \hat{I}(\beta > 0)$ and the monostatic receiver $\hat{I}_{\mathrm{mono}}(\beta_v = 0.003°) \approx I(0)$ (Sect. 2.2.2). This approximation is well justified considering that the expected width of the peak is at least one order of magnitude larger than the small bistatic angle of the monostatic receiver (cf. Fig. 6) and that rounding of the peak tip due to weak absorption can be expected. The TanDEM-X measurement can, therefore, be described by Eq. (8) as:

$$I_{r,0}(\beta) = \frac{I(\beta)}{I(0)} = \frac{I_0(1 + B_C(\beta))}{I_0(1 + B_C(0))} = \frac{1 + B_C(\beta)}{1 + B_C(0)} \tag{13}$$

The intensity ratio $I_{r,0}(\beta)$ is 1.0 for $\beta = 0$ and reaches its minimum 0.5 at $\beta \to \infty$ when absorption is negligible. With increasing absorption the contrast $I(\beta)/I(0)$ lowers and $I_{r,0}(\beta \to \infty)$ increases from 0.5 to eventually 1.0 when the CBOE is negligible.

A lower limit for the enhancement $B_C(\beta = 0)$ can be quickly estimated: because $B_C(\beta_{\mathrm{max}}) > B_C(\infty) = 0$ it follows from Eq. (13) that

$$B_C > I_{r,0}^{-1}(\beta_{\mathrm{max}}) - 1 = \frac{\hat{I}_{\mathrm{mono}}}{\hat{I}_{\mathrm{bist}}(\beta_{\mathrm{max}})} - 1, \tag{14}$$

i.e. the enhancement $B_C$ is at least as large as the relative difference between the monostatic and the bistatic backscatter, $\hat{I}_{\mathrm{mono}}$ and $\hat{I}_{\mathrm{bist}}$, at the largest available bistatic angle $\beta_{\mathrm{max}}$.

To fit the model, we used winter data from the accumulation area (Sect. 2.2.1). To determine the optimal value of the parameter pair $(\Lambda_T, \Lambda_A)$, and the 95% confidence intervals, we used the TRF method (Sect. 2.3.1), and set the starting parameter of $(\Lambda_T, \Lambda_A)$ to $(2\,\mathrm{m}, 20\,\mathrm{m})$. However, a sampling of the RMSE$[\hat{I}_{r,0}(\beta) - I_{r,0}(\beta)]$ in the parameter space of $\Lambda_T, \Lambda_A$ around the optimal value revealed that the global minimum is weakly constrained, and solutions across a large span of values of $\Lambda_A$ provide an acceptably low RMSE value. Thus, to explore multiple parameter pair values, we sampled a range of values of $\Lambda_A$, and used a downhill simplex method implemented in the `amoeba` IDL function (Nelder and Mead, 1965) to determine the corresponding $\Lambda_T$.

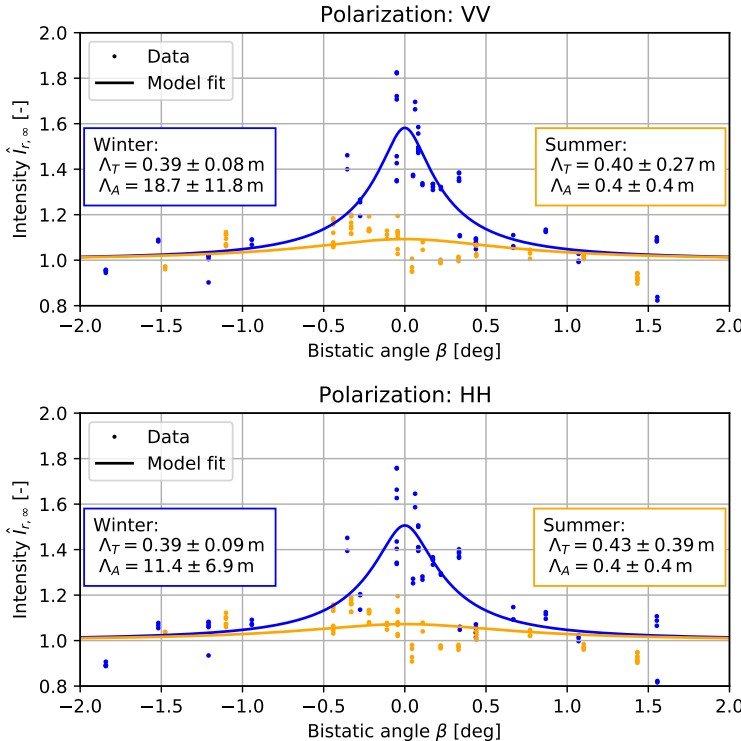

**Figure 7.** Intensity ratio $\hat{I}_{\mathrm{r},\infty}(\beta)$ observed during the ground-based experiment (KAPRI), showing backscatter enhancement in the winter dataset (blue). For the summer dataset (orange), the comparable values of the $\Lambda_T$ and $\Lambda_A$ estimates, as well as their relatively large confidence intervals, indicate that the CBOE peak was not detectable. The observed intensities $\hat{I}(\beta)$ were averaged over the whole region of interest, for both polarizations, VV and HH. The blue and orange lines describe the least-squares fit model defined by eqs. (6), (7), and (11) for winter and summer respectively. The colored box for each dataset shows the best-fit value, and the 95% confidence interval for the model parameters $\Lambda_T, \Lambda_A$ describing the scattering and absorption mean free paths.

## 3   Results

### 3.1   Ground-based observations - KAPRI

Figure 7 shows the observed intensity ratio $\hat{I}_{\mathrm{r},\infty}(\beta)$ defined by Eq. (12) at HH and VV polarization, and the least-squares best fit of the model defined by Eqs. (6), (7), and (11).

For the winter dataset a clear intensity peak is detected, with a HWHM of approx. $0.25°$ and amplitude $B_C(0) \approx 0.5\,(1.8\,\mathrm{dB})$, corresponding to $\Lambda_T \approx (0.4 \pm 0.1)\,\mathrm{m}$ for the HH and VV polarization. The derived absorption lengths $\Lambda_A$ are much longer than the scattering lengths with $\Lambda_A \approx (11 \pm 7)\,\mathrm{m}$ for the HH polarization and $\Lambda_A \approx (19 \pm 12)\,\mathrm{m}$ for the VV polarization. For the 390 summer dataset, the flat profile of the observed intensity curve indicates that very little or no backscatter enhancement is present – this is reflected in the model fit in the low value and large confidence interval (relative to the value) of the absorption

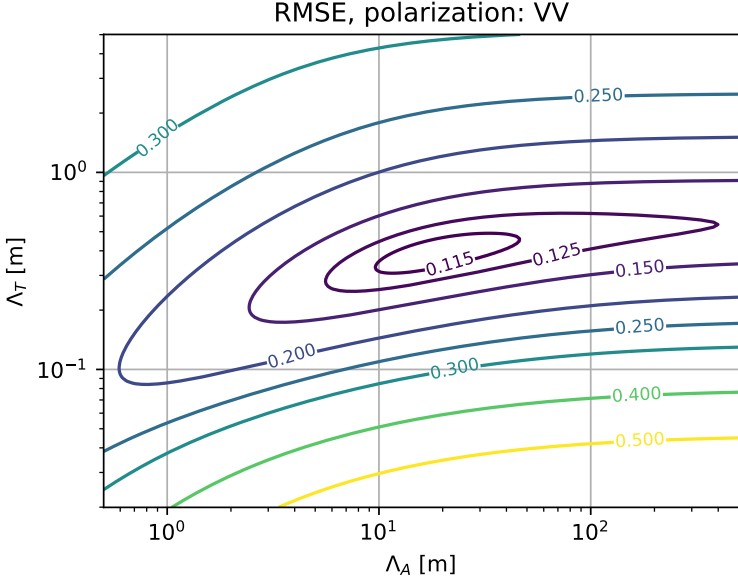

**Figure 8.** Contour plot of root mean square error (RMSE) between measured and modeled winter data in Ku-band (Fig. 7) for different parameter pairs $\Lambda_A, \Lambda_T$. The plot indicates a clear global minimum because the CBOE peak height, and hence $\Lambda_A$, can be well estimated due to the availability of ground-based KAPRI measurements at large bistatic angles $\beta > 1°$.

length $\Lambda_A \approx (0.4 \pm 0.4)\,\text{m}$. The estimates of the scattering length in the summer dataset ($\Lambda_T \approx (0.40 \pm 0.27)\,\text{m}$ and $\Lambda_T \approx (0.43 \pm 0.39)\,\text{m}$ for the VV and HH polarization respectively) have comparable value to the winter estimates, however the much larger confidence intervals indicate that the value of $\Lambda_T$ could not be determined more precisely for the summer dataset due to the absence of a clear enhancement peak. The uncertainty of the value estimates corresponds to the 95% confidence interval.

Figure 8 shows the RMSE of the model fit to the dataset with VV polarization, dependent on values of the fit parameters $\Lambda_T, \Lambda_A$. A clear global minimum can be found in the 2-dimensional parameter space at the best-fit parameter values mentioned above, with RMSE of approximately $0.11$. Residuals of the model fits for selected values of parameters $\Lambda_T, \Lambda_A$ are shown in supplementary Fig. S1.

## 3.2 Satellite observations - TanDEM-X

### 3.2.1 Jungfrau-Aletsch region

The large number and wide coverage of the TanDEM-X scenes allow an analysis of the dependency of $\hat{I}_{r,0}$ on $\beta$ for different land cover types. In Fig. 9, where the color of data points refers to $I_{\text{mono}}$ (Fig. 10) to distinguish between dry and wet snow, only $\hat{I}_{r,0}$ in the accumulation area $> 3500\,\text{m}$ (a, b) shows a significant dependence on $\beta$: the ratio $\hat{I}_{r,0}(\beta)$ forms a clear peak

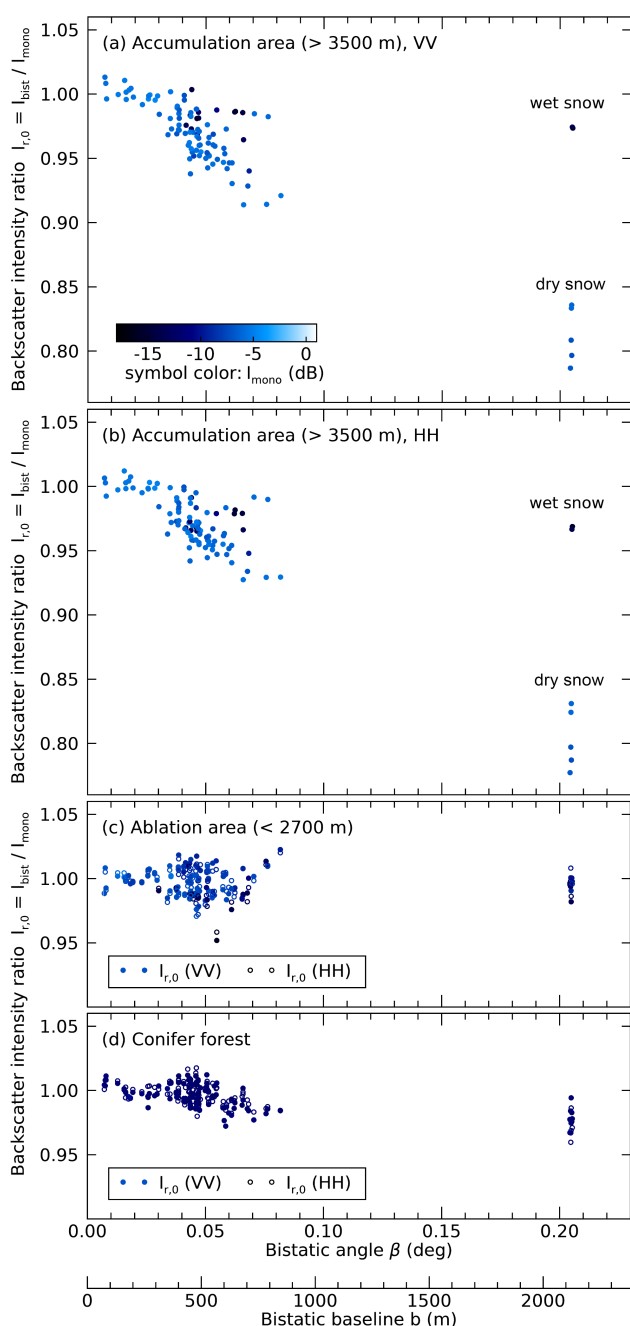

**Figure 9.** (a,b) The intensity ratio $\hat{I}_{r,0}$ observed by the satellite TanDEM-X in the accumulation area. Backscatter enhancement is indicated by the significant dependence of $\hat{I}_{r,0}$ on the bistatic angle $\beta$ for dry-snow observations at both polarizations (VV, HH). The symbol color indicates the monostatically measured radar brightness $I_{mono}$ and helps to differentiate between wet snow (dark blue) and dry snow (light blue), see Fig. 10(a). (c): The ablation area of Great Aletsch Glacier, located below 2700 m, is covered by $0 - 3\,\mathrm{m}$ of snow in winter but does not show any dependence of $\hat{I}_{r,0}$ on $\beta$. (d) areas covered by conifer forest show no significant dependence on $\beta$.

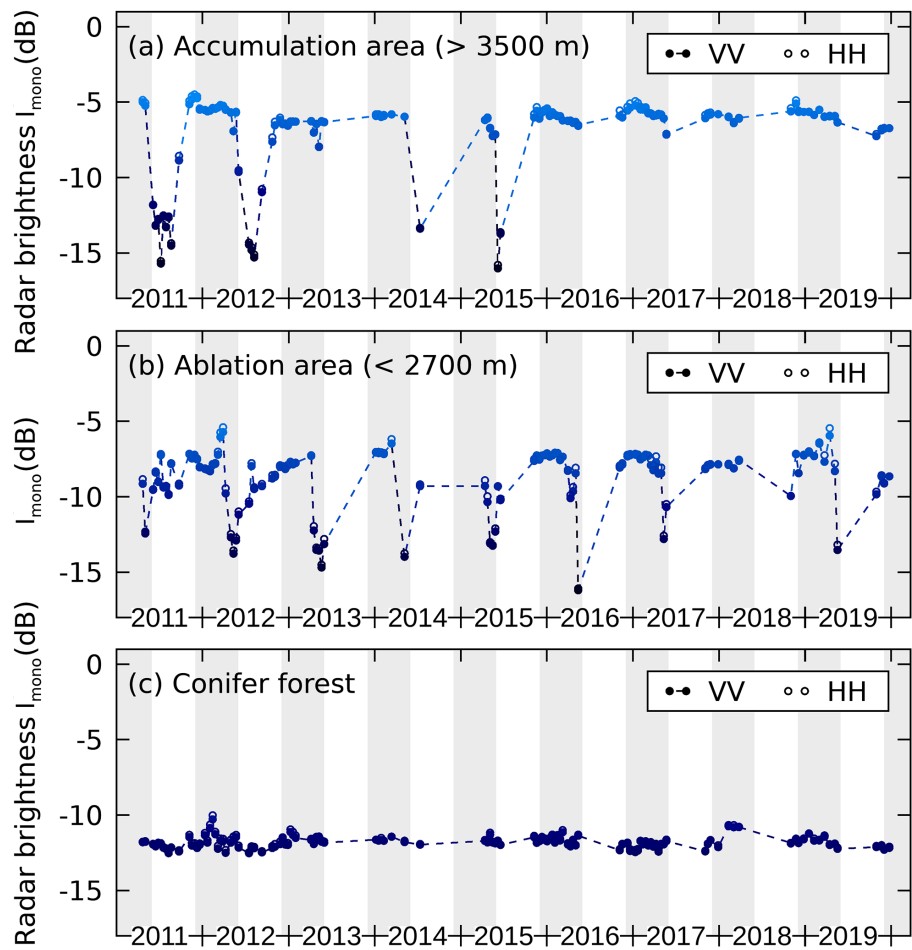

**Figure 10.** Time series of the monostatically measured radar brightness $\hat{I}_{mono}$ observed by the satellite TanDEM-X for three different land cover types: (a) deep firn in accumulation area above 3500 m, (b) the tongue of Great Aletsch Glacier which is covered by 0-3 m of snow in winter. In (a) and (b) seasonal variations of $I_{mono}$ provide a good indicator to distinguish between dry and wet snow. Gray shading indicates the period 01 December–31 May when dry firn is most likely present in the accumulation zone. This period was selected for estimation of the absorption and scattering lengths $\Lambda_A$ and $\Lambda_T$. A backscatter difference between the VV and HH polarized data is hardly visible. (c) Compared to snow, forest (mainly conifer forest) shows only small variations of the radar brightness when summer and winter acquisitions are available (2011–2013).

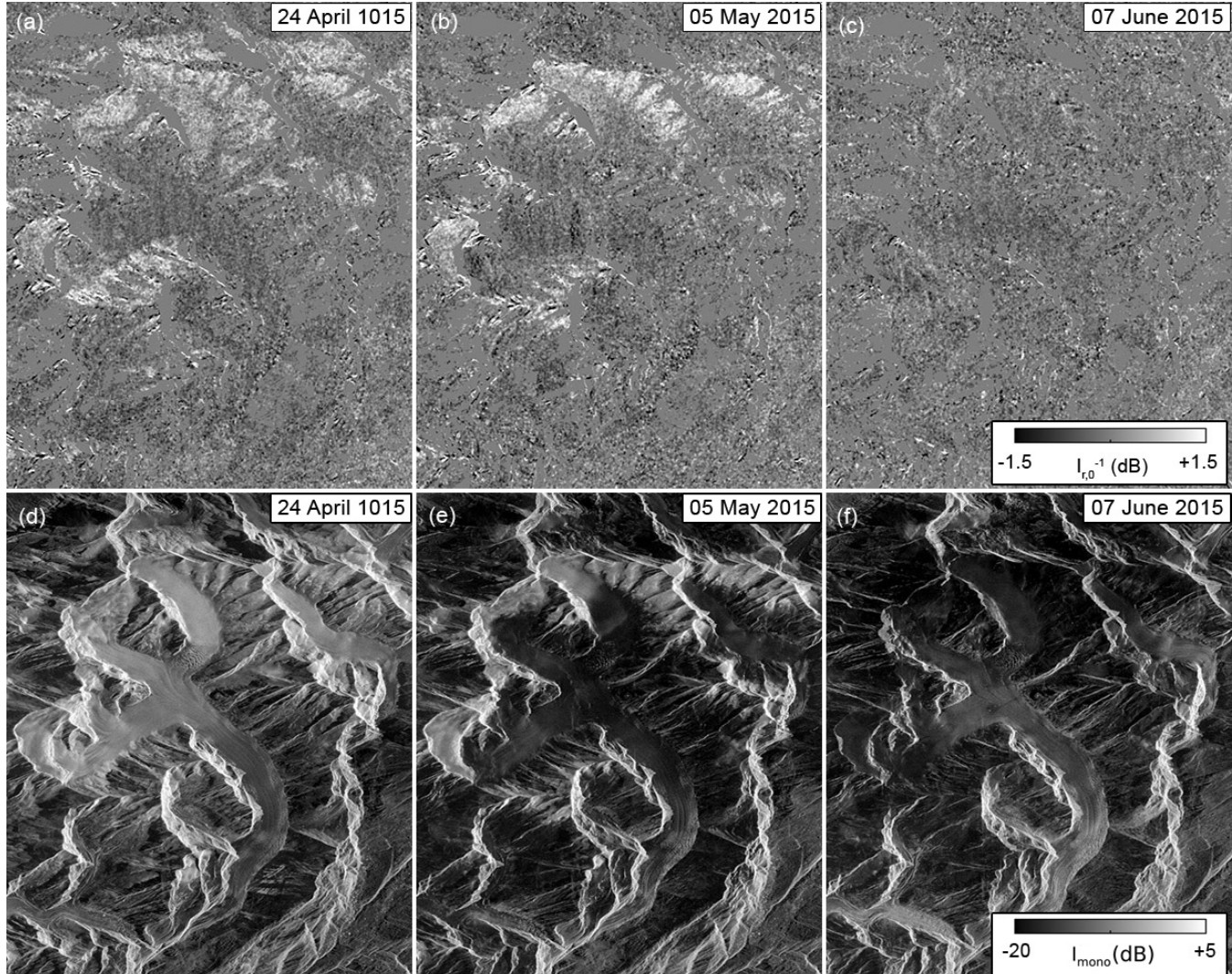

**Figure 11.** (a-c): monostatic-to-bistatic backscatter ratio $\hat{I}_{r,0}^{-1}$, observed by TanDEM-X at the largest available bistatic angles $\beta_{max} = 0.2°$ before (left) and during snow melt (middle, right). (d-f): radar brightness for the same dates. Areas covered by wet snow appear dark. Great Aletsch Glacier is flowing clockwise from top to bottom. In (a, b), high altitude areas (above 3000 m and above 3400 m) show backscatter enhancement. In (c) wet snow is present in the entire scene and absorption prevents the CBOE. In (f) an increase of the backscatter intensity becomes visible on the tongue of Great Aletsch Glacier and on nearby vegetation covered slopes, indicating that in these areas all snow has melted. Images are shown in slant range/azimuth coordinates.

that shows some rounding between $\beta = 0$ and $0.05°$, characteristic of weak absorption. At both polarizations, VV and HH, and only at dry snow conditions, at the largest available bistatic angles $\beta_{max} = 0.2°$ the bistatic backscatter intensity $\hat{I}_{bist}$ is reduced by approximately 20% compared to $\hat{I}_{mono}$. For wet snow (dark dots) no reduction of $\hat{I}_{r,0}$ is observed at $\beta_{max}$. In contrast,

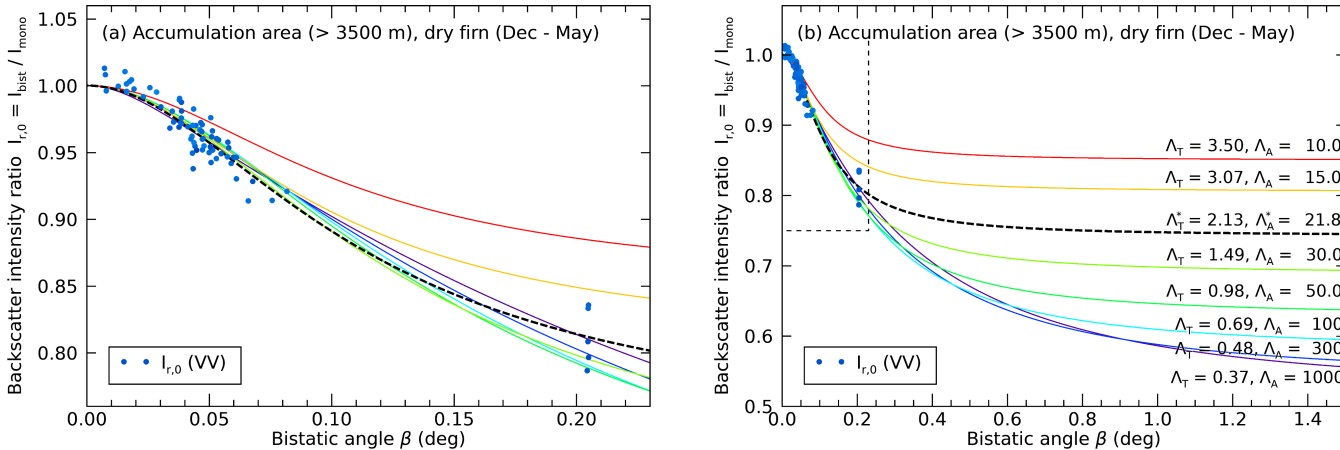

**Figure 12.** (a): Backscatter ratios $\hat{I}_{r,0}(\beta)$ (dots) restricted to dry firn observations in winter (01 December–31 May; $I_{\mathrm{mono}} > -8\,\mathrm{dB}$) in the accumulation areas of the Jungfrau-Aletsch region from TanDEM-X at VV polarization. Colored lines indicate different CBOE model curves, Eq. (13), parametrized by a range of scattering and absorption parameter combinations $(\Lambda_T, \Lambda_A)$ that can describe the data to different degrees. (b): same as (a) (in dashed box) but zoomed out to visualize at which $\beta$ the different model fits converge to the flat region of the intensity curve $I_{\mathrm{r,0}}(\beta \to \infty)$ where coherent backscatter enhancement is negligible. Measurements at larger bistatic angles $\beta > 0.5°$ could substantially better constrain the total peak height (and thereby $\Lambda_A$) but such large angles are currently not available.

neither the ablation area of Great Aletsch Glacier, Fig. 9(c), nor areas covered by conifer forest, Fig. 9(d), show any significant

dependence of $\hat{I}_{\mathrm{r,0}}$ on $\beta$.

To investigate the spatial distribution of areas that show enhanced backscattering, Fig. 11 shows imagery of the monostatic-to-bistatic backscatter ratio $\hat{I}_{\mathrm{r,0}}^{-1}$ together with the radar brightness $I_{\mathrm{mono}}$ for a series of three acquisitions with $\beta = \beta_{\mathrm{max}}$ at the onset of snow melt in April/May 2015: On 24 April (Fig. 11a, d), backscatter enhancement is visible for a considerable amount of the area, corresponding to glaciers at high altitude ($> 3000\,\mathrm{m}$). On 05 May (Fig. 11b, e), the backscatter enhancement is

limited to high altitudes, because snow melt is occurring up to an altitude of approximately 3300 m. On 07 June (Fig. 11c, f), snow melt reaches the peaks of the highest mountains (4274 m) and no enhanced backscattering is detectable at any place.

To estimate the scattering and absorption parameters $\Lambda_T$ and $\Lambda_A$, as well as the peak width and the backscatter enhancement, we fitted the model, Eq. (13), to the dry firn data of the accumulation area, constrained to winter acquisitions (the selection is specified in Sect. 2.3.2 and indicated by gray shading in Fig. 10a). Figure 12(a, b) shows the selected dataset at two different

scales of $\beta$. The solution with the minimal RMSE value is indicated by the black dashed line. This solution corresponds to $\Lambda_T = 2.16 \pm 0.36\,\mathrm{m}$ and $\Lambda_A = 21.77 \pm 2.72\,\mathrm{m}$ (95% confidence interval, VV polarization), an enhancement of $B_C = 35\%$ (+1.3 dB), and a HWHM of the peak of $0.12°$. From the HH polarized data (not shown) we obtained $\Lambda_T = 1.62 \pm 0.35\,\mathrm{m}$ and $\Lambda_A = 25.88 \pm 5.27\,\mathrm{m}$, corresponding to $B_C = 41\%$ (+1.5 dB), and a HWHM of $0.14°$.

Figure 12(a) also illustrates that the available data, sampled with a limited range of bistatic angles, allow multiple model

solutions with similar RMSE values (colored lines). The solutions are parametrized by different pairs of scattering and absorp-

| $\Lambda_A$ (m) | $\Lambda_T$ (m) | $B_C(0)$ | HWHM (deg) | RMSE |
|---|---|---|---|---|
| 1000 | 0.37 | 0.92 | 0.28 | 0.0111 |
| 300 | 0.48 | 0.85 | 0.25 | 0.0117 |
| 100 | 0.69 | 0.72 | 0.21 | 0.0129 |
| 50 | 0.98 | 0.59 | 0.17 | 0.0129 |
| 30 | 1.49 | 0.45 | 0.14 | 0.0116 |
| **25.9** | **1.63** | **0.41** | **0.14** | **0.0103** (HH) |
| **21.8** | **2.13** | **0.35** | **0.12** | **0.0106** (VV) |
| 15 | 3.08 | 0.24 | 0.10 | 0.0145 |
| 10 | 3.50 | 0.18 | 0.11 | 0.0257 |

**Table 1.** Scattering length $\Lambda_T$, peak height $B_C(0)$, and peak width (HWHM) for a set of chosen absorption lengths $\Lambda_A$ determined from the TanDEM-X dataset (VV) of dry firn in the high altitude accumulation area. The bold lines indicate the optimal parameter pair at VV and HH polarization. RMSE is the root mean square error between the measured and the modeled value of $I_{r,0}$.

tion mean free paths $\Lambda_T, \Lambda_A$ and were determined by finding $\Lambda_T$ for a fixed $\Lambda_A$. Table 1 summarizes these parameter pairs and lists for each solution the modeled peak characteristics and the RMSE with respect to the measured data.

Figure 12(b) illustrates how the different model solutions asymptotically reach the incoherent background $I_{r,0}(\beta \to \infty)$ at large bistatic angles. The sampling of larger bistatic angles would reveal whether significantly lower values of $I_{r,0}(\beta)$ than
observed exist, and would, therefore, allow a better constraint of the model parameters.

A contour map of the RMSE between the measured and the modeled values (VV) is shown in Figure 13. While the shallow global minimum (RMSE = 0.0106) is located at the optimal solution $\Lambda_T = 2.12\,\mathrm{m}, \Lambda_A = 21.8\,\mathrm{m}$, multiple other solutions exist, that show only slightly higher RMSE values between 0.011 and 0.015 (see also Table 1). This set of possible parameter pairs $(\Lambda_A, \Lambda_T)$ forms a non-linear curve (1-dimensional manifold) in the 2-dimensional parameter space (red "+"-symbols in
Figure 13).

## 4   Discussion

### 4.1   Cause of enhancement: CBOE vs. SHOE

In both Ku-band and X-band observations of terrestrial snow, we observed narrow intensity peaks with an angular width of a fraction of $1°$. These peaks are clearly attributable to the CBOE as opposed to the SHOE, which follows from a comparison of
the effects' properties as described by Hapke (2012, Chapter 9):

Firstly, the SHOE requires that the scatterers are much larger than the wavelength of the incident radiation so that they can cast sharp shadows. This requirement can hardly be fulfilled at radar wavelengths of several centimeters, since ice particles in snow have average diameters on the order of millimeters (Kuga et al., 1991). A grain size of $0.3 - 1.5\,\mathrm{mm}$ was observed in the seasonal snowpack studied by the KAPRI experiment (Fig. 2). The snowpack did not show any cm-sized ice structures

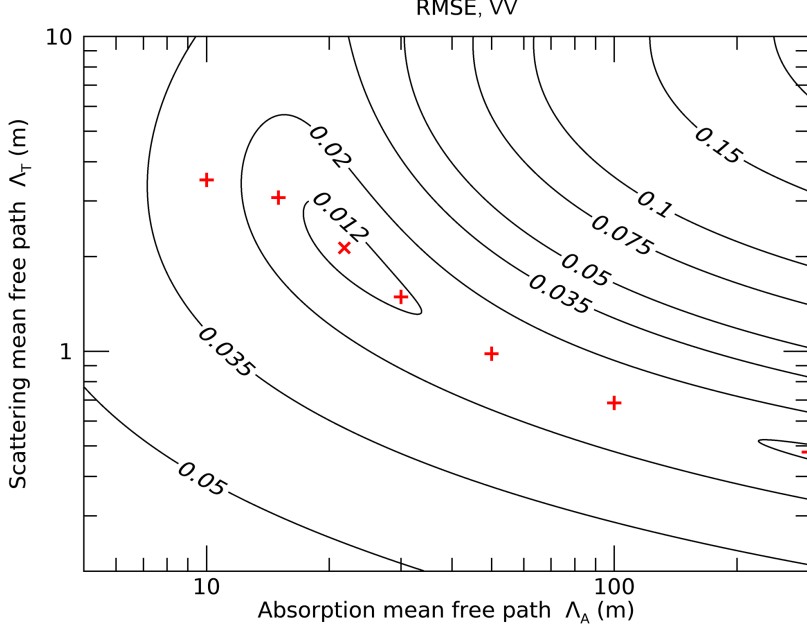

**Figure 13.** Contour map of root mean square error (RMSE) between measured and modeled data for TanDEM-X (Fig. 12) for different pairs of $\Lambda_A, \Lambda_T$. The plot shows a weak global minimum because acquisitions at sufficiently large $\beta$, that could better constrain $\Lambda_A$, were not currently available. The model parameters and RMSE values for the red "+" symbols are given in Table 1. For these points, the parameter value of $\Lambda_T$ was estimated by nonlinear least squares minimization for different choices of $\Lambda_A$.

that could have been caused by strong melt-events. Furthermore, the narrow peak width is in agreement with characteristics of the CBOE, while a SHOE peak usually has a width of several degrees or tens of degrees, depending on particle size distribution. Finally, SHOE is only present in media where single scattering is dominant. Multiple scattering processes decrease the amplitude of the SHOE and increase amplitude of the CBOE. Since dry snow is a weakly absorbing medium for microwaves where multiple (i.e. volume) scattering is considerable (Kuga et al., 1991), the CBOE is expected to be the dominant effect.

## 4.2 Observations of CBOE

### 4.2.1 Ground-based observations - KAPRI

Figure 7 shows a statistically significant enhancement peak for the winter acquisition, and a lack of such a peak for the summer dataset, which was acquired using an identical target region of interest, identical acquisition procedure (except platform substitution to allow movement on snow/road), and identical processing pipeline. The summer dataset thus serves as a useful control which ensures that the detected enhancement peak is not an erroneous artifact of the bistatic data processing pipeline, and it also indicates that the enhancement peak is indeed caused by the snow layer present on the hillside. In the summer scenario (i.e. absence of a clear backscatter enhancement peak), the model, described in Sect. 2.3 and visualized in Fig. 6,

predicts that the absorption length is shorter or equal to the scattering length ($\Lambda_A \leq \Lambda_T$). An interpretation of this scenario is that higher-order scattering paths are suppressed due to absorption, and thus the summer scenario is dominated by a single-scattering process. In the summer scenario the model becomes much less sensitive to the precise value of $\Lambda_T$ (which is a measure of the width of the peak), and thus estimates of this value have much higher uncertainty as opposed to the case of a clearly detectable enhancement peak in winter.

The best-fit values of model parameters in Fig. 7 indicate a scattering mean free path value $\Lambda_T$ between $30$ and $50\,\mathrm{cm}$, and an absorption length $\Lambda_A$ between $6$ and $24\,\mathrm{m}$ at both polarizations which matches well the observations by Wiesmann et al. (1998). No statistically significant difference of the parameter estimates is observed between the HH and VV polarized data. This is well-aligned with the theoretical model of Mishchenko (1992b), in which there is only a very small difference between the co-polarized backscatter enhancement factors at these two polarizations. The HWHM of the angular peak of $\approx 0.25°$ is sufficiently wide so that KAPRI's transmit antennas' non-zero size, which limits the angular resolution to $0.05°$ (Sect. 2.1.2), has only a very limited effect on the precision with which the width and height of the peak can be determined.

The snow depth during the winter acquisitions was measured on-site as approximately $1.5\,\mathrm{m}$ (Fig. 2), and thus the estimate of $\Lambda_A$ is several times higher than the snow depth. The extremely shallow local incidence angle (above $70°$ for the vast majority of the ROI area) and short transport mean free path $\Lambda_T$ would likely lead to longer trajectories of the radiation through the snow medium before reaching the ground. Nevertheless, the optical thickness $\tau_d = E\,d \approx d/\Lambda_T$ of the snow depth $d$ of only 3–4 scattering mean free paths $\Lambda_T$ could limit higher order scattering. While Tsang and Ishimaru (1985) conclude that already at $\tau_d = 4$ models approximate well the half-space solution (where $\tau_d = \infty$), Van Der Mark et al. (1988, Figs. 9,12) show that the peak height and width, at least for very weakly absorbing media ($\Lambda_A \gg \Lambda_T$), might be affected up to $\tau_d \approx 30$. Missing higher order scattering, in turn, is an explanation for a rounding of the peak shape (Akkermans et al., 1988, Fig. 7). During the field experiment, a corner reflector lowered to the bottom of a $1.55\,\mathrm{m}$ deep snow pit with vertical walls was still visible, indicating that at least a fraction of microwaves reached the ground, thereby limiting higher order scattering.

### 4.2.2 Satellite observations - TanDEM-X

A significant dependence of the backscatter intensity on the bistatic angle, $\hat{I}_{\mathrm{r},0}(\beta)$, is only visible in the accumulation zones of the Jungfrau-Aletsch region with altitude $H > 3500\,\mathrm{m}$ (Fig. 9a, b). Above this altitude, a firn layer with below freezing snow temperatures is present to a depth of several tens of meters (Haeberli and Alean, 1985; Suter et al., 2001). This thick and cold firn layer represents a disordered medium where multiple scattering is possible and at the same time microwave absorption is weak, because liquid water is absent. The existence of the CBOE in dry firn is further supported by the spatial and temporal distribution of an enhanced brightness ratio $\hat{I}_{\mathrm{r},0}$. Spatially, the enhancement matches to the accumulation area of high altitude glaciers in the Jungfrau-Aletsch region (Fig. 11). Temporally, the backscatter enhancement vanishes in these areas when snow melt sets in and thus the scattering predominantly takes place at the snow surface. These observations were confirmed by additional data from the Teram-Shehr/Rimo Glacier in the Karakorum (see supplements).

The rounding of the peak shape in Fig. 12(a) indicates that either absorption or a limited thickness of firn is present in the accumulation area. Field measurements indicate cold firn in at least the upper $8\,\mathrm{m}$ (Bannwart, 2021) and literature data

indicate that temperate firn might be present at around 15 m below the surface (Suter et al., 2001). The global minimum at $\Lambda_T = 2.1\,\text{m}, \Lambda_A = 21.8\,\text{m}$ in Fig. 13 might therefore provide a realistic estimate for $\Lambda_A$ as larger absorption lengths are difficult to conform to field measurements. Our observed values for absorption and scattering lengths also agree with the measurements by (Wiesmann et al., 1998).

On the tongue of Great Aletsch Glacier, where a seasonal snowpack is present during winter, no backscatter enhancement was observed in X-band (Fig. 9c). As seasonal snow is younger than multi-year firn, smaller snow grain sizes are expected, resulting in scattering lengths larger than the value $\Lambda_T = 2.1\,\text{m}$ determined for the accumulation area. The thickness of the seasonal snowpack of 0–3 m corresponds therefore to an optical thickness $\tau_d \approx 1$ or less, which considerably affects the peak intensity (Van Der Mark et al., 1988, Fig. 9). In consequence, the single scattering at the (possibly rough) snow-ice interface at the bottom of the snowpack can remain the dominant scattering process. The low average number of scattering events in the seasonal snow volume is, therefore, not sufficient for the CBOE to occur on the ablation area of Great Aletsch Glacier.

In forest covered areas, no significant dependency of $\hat{I}_{\text{r,0}}$ on $\beta$ is visible. We think the reason is that, compared to dry snow, multiple scattering at X-band is reduced in forest due to absorption of microwaves, hence the CBOE is prevented.

We also did not observe coherent backscatter enhancement in any area other than the high accumulation area, even though the tongue of Aletsch glacier is highly crevassed and valley slopes are covered by rock debris. From this we conclude that in the X-band, rough surfaces do not elicit the CBOE.

### 4.2.3 Comparison of bistatic measurement geometries

The KAPRI experiment sampled a larger range of bistatic angles (up to $1.92°$) so that the flat incoherent intensity background $I_0$ could be sampled. Therefore, both the width and the height of the enhancement peak in winter can be constrained much better than with the TanDEM-X observations where $\beta_{\text{max}} \approx 0.2°$. This, in turn, translates to better constrained estimates of parameters $\Lambda_T$ and $\Lambda_A$ as illustrated by the clearly visible global minimum in the plot of the RMSE value in Fig. 8 as compared to Fig. 13.

Compared to the KAPRI experiment, the bistatic angles sampled by TanDEM-X are relatively small, making it possible that not the entire peak of the CBOE was sampled. In consequence, the bistatic data at $\beta_{\text{max}} = 0.2°$ might still be affected by the CBOE. Missing measurements at larger bistatic angles result in a weak constraint of the parameter pair $(\Lambda_T, \Lambda_A)$, permitting a range of value pairs that each can fit the data (Table 1 and Figs. 12 and 13). To better constrain the observed values of $\Lambda_T, \Lambda_A$, bistatic angles of at least $\beta = 0.5°$ would need to be sampled by TanDEM-X. However, such larger angles are currently not available.

### 4.3 Impact of the CBOE on backscatter observations

Generally, the existence of a narrow backscatter enhancement peak around the monostatic direction needs to be kept in mind when performing backscatter measurements of snow, regardless of whether the used sensors are considered monostatic or bistatic. On one hand, for truly monostatic sensors the CBOE is strongest. On the other hand, some radar sensors are considered as monostatic even though they have a small but non-zero spatial separation between the transmitting and receiving antennas. Due to this bistatic baseline, the detected backscatter intensity value could be significantly reduced compared to the value

that would have been detected by a truly monostatic sensor. When prior estimates of $\Lambda_T$ and $\Lambda_A$ over a particular medium are available, Eq. (6) could be used to roughly estimate the width and height of the peak, which can subsequently be used to estimate bistatic angle values where the CBOE might affects the measurements (see also Sect. 4.4).

As an example of the necessity to precisely align the measurement geometry to the expected width of the peak, Tan et al. (2015) compared modeled results to active and passive microwave measurements at X- to Ka-band performed in Finland, Sodankylae as part of the NoSREx field experiment (Lemmetyinen et al., 2016). The active measurements were performed with the SnowScat instrument (Wiesmann et al., 2010). The model, which includes backscatter enhancement into the DMRT theory, might provide a significant step forward for modeling of the radar backscatter signal from snow. However, if the peak width of the CBOE in the NoSREx experiment is comparable to the narrow observed peak widths in our study (HWHM $\approx 0.2\pm0.1°$), the bistatic angles of the SnowScat measurements would actually be one order of magnitude too large to observe the CBOE. This follows from the instrument height of 9.6 m, the incidence angle range of 30–60° (Lemmetyinen et al., 2016), and the antenna separation of 72 cm (Wiesmann et al., 2010; Wiesmann and Werner, 2010), resulting in bistatic angle values $\beta = 1.8 - 3.7°$.

Except for possible extreme cases of a medium causing an extremely narrow enhancement peak (with width on the order of thousandths of a degree), the velocity-induced bistatic angle $\beta_v$ of moving radar platforms is negligible in the context of the CBOE. For the side-looking geometry, the bistatic angle $\beta_v$ caused by platform motion with velocity $v$ can be calculated as $\beta_v = 2v/c$, where $c$ is the speed of light. Thus, for all conventional sensors – even for satellite platforms in low Earth orbit moving at speeds of 6–8 km s$^{-1}$ – the resulting value of $\beta_v$ is on the order of thousandths of a degree or less.

## 4.4 Link to the microstructure of snow

In the model outlined in Sect. 2.3 and which assumes an optically thick medium, the two parameters $\Lambda_T$ and $\Lambda_A$, defining the peak shape, can be linked to the snow microstructure and to the density of snow. The transport mean free path $\Lambda_T \geq \Lambda_S$ is a measure of the medium's scattering properties, and corresponds to the scattering mean free path $\Lambda_S = S^{-1}$ for particles that scatter EM radiation symmetrically in the forward and backward direction (Hapke, 2012, Eq. 7.24b and Sect.5.2.7), see also (Van Der Mark et al., 1988, Sect. IV-A). For negligible absorption, $\Lambda_T$ describes the one-way penetration depth where the incident radiation is reduced to $1/e$ by side-way scattering. While in the context of CBOE modelling, the (volume-averaged) scattering coefficient $S$ is derived from the scattering cross section of individual particles in sparse media (Hapke, 2012, Ch. 5), (Ishimaru, 1978, Ch. 2) and (Van Der Mark et al., 1988), for snow the scattering coefficient needs to be estimated with dense-media radiative transfer theories like DMRT (e.g. (Tsang et al., 2007)) or IBA (Mätzler, 1998) as shown in [Sect. 3.1](Picard et al., 2018) and as already indicated by (Mishchenko, 1992a). The description of the SMRT model (Picard et al., 2018) provides a direct relation between the scattering coefficient $S$ and the phase function and links these to the autocorrelation function of the mediums indicator function that represents the spatial 3D microstructure of the snow/ice matrix (Löwe and Picard, 2015). Still, both theories, DMRT and IBA, are not yet sufficiently parametrized by field-measurable quantities (Picard et al., 2018). An empirical relation to link the microstructure to $S$ is given in (Wiesmann and Mätzler, 1999).

The absorption mean free path $\Lambda_A = A^{-1}$ is a measure of the medium's absorbing properties given by the volume-averaged absorption coefficient $A$ (Hapke, 2012, Eq. 7.18a). For negligible scattering, $\Lambda_A$ describes the absorption length where the

incident radiation intensity is reduced to $1/e$; For continuous media (without scatterers) $A$ would be equivalent to the absorption coefficient $\alpha = 4\pi/\lambda n_i$ with $n_i$ the imaginary part of the refractive index. For snow, Picard et al. (2018) recommends computation of $n_i$ from the Polder-van Santen formular, e.g. in (Sihvola, 2000; Mätzler, 1998; Wiesmann and Mätzler, 1999); the refractive index of pure ice is given by Warren and Brandt (2008).

The absorption and Scattering coefficient sum up to the extinction coefficient $E = S + A$ which corresponds in the sparse-media models from Tsang and Ishimaru (1984, 1985) and Van Der Mark et al. (1988, below Eq.(26b)) by $E = 2K''$ to the effective propagation constant $K''$ that is related to particles absorption and scattering properties described by the scattering amplitude $f$.

## 4.5 Limitations of the model

The CBOE model used in this work can accurately predict the peak shapes observed in various volume-fractions of colloidal suspensions where the particle sizes are within the order of magnitude of the wavelength (Van Der Mark et al., 1988; Akkermans et al., 1988). The parameters of the model, in particular the scattering coefficient $S \propto \Lambda_T^{-1}$ and absorption coefficient $A = \Lambda_A^{-1}$ that determine the shape of the CBOE peak, can, in theory, be estimated from the microstructure and density of the snow pack when considering dense-media scattering theories (Picard et al., 2018), see also (Mishchenko, 1992a) who addresses already a snow-like structure. However, the often complex (multi-layer) snow structure (e.g., Proksch et al. (2015)) together with current limitations in accurately predicting the scattering coefficients from the snow microstructure (Vargel et al., 2020) might prevent a precise estimate of the peak shape even though a rough estimate of the peak width is feasible. An additional limitation for an accurate estimation of $\Lambda_A$, possibly also $\Lambda_T$, results from the assumption that the scattering medium fills a semi-infinite space whereas the snow pack has a limited optical thickness $\tau_d$. Hence, $\Lambda_A$ might be underestimated due to limited layer thickness (Van Der Mark et al., 1988; Van Albada et al., 1988).

Our observations of the CBOE peak shape originate from a natural (non-homogeneous) snow cover and currently no laboratory experiments of the CBOE at microwave frequencies, including a precise characterization of the microstructure, are available. Such experiments could validate the used model and might indicate whether adaption of the model or introduction of additional correction factors could be required in order to precisely link the microstructure to the CBOE peak shape. Nevertheless, we clearly observed the CBOE peak in natural snow, which can lead to development of new methods for snow and ice monitoring.

## 4.6 Applications based on the CBOE

In TanDEM-X data we have observed a backscatter enhancement of at least $1.3\,\mathrm{dB}$ for firn covered areas of the European Alps and in the Karakorum, while for firn free areas no backscatter enhancement could be observed. This suggests that detection of deep firn with X-band is possible when large enough bistatic angles $\beta > 0.2°$ are available.

For seasonal snow, we observed a clear CBOE peak ($\sim 1.8\,\mathrm{dB}$ of backscatter enhancement) at Ku-band using KAPRI whereas in X-band we were not able to observe an enhancement. The higher sensitivity of high-frequency systems (Ku- or possibly Ka-band) to detect the CBOE in seasonal snow results from the shorter scattering length, since sufficiently high-

order scattering events can occur within the snow layer of limited thickness. Furthermore, the difference between KAPRI observations in summer (no CBOE from vegetation) and winter (snow-induced CBOE) demonstrates how Ku-band bistatic observations could potentially provide a mean to discriminate dry snow from vegetation and therefore provide a mean to map snow cover extent. Beyond that, the frequency dependency of the scattering lengths makes a characterization of the intensity of the CBOE at multiple frequencies possible which, in turn, could allow a quantitative characterization of the height or water equivalent of seasonal snow. The area covered by snow, the snow depth, and the snow-water equivalent are considered key data products for the snow Essential Climate Variable (Belward, 2016). Bistatic missions characterizing the CBOE occurring in snow can thus be an asset in mapping these data products.

In terms of polarimetric measurements, the results of this study, as well as experimental work and theoretical models (van Albada et al., 1987; Mishchenko, 1992b, c; Wolf and Maret, 1985), indicate that the effect is present predominantly in co-polarized channels, and the effect is equally strong at both horizontal and vertical polarizations. Nevertheless, in further studies the use of full-polarimetric radar systems can still be advantageous, e.g. to decisively differentiate the CBOE and the SHOE based on their different impact on linear and circular polarization ratios (Rignot, 1995; Hapke, 2012, Sect. 9.4).

Existing bistatic ground-based SAR sensors (Pieraccini and Miccinesi, 2017; Wang et al., 2019), and to a certain extent also airborne bistatic SAR sensors (Dubois-Fernandez et al., 2006; Meta et al., 2018) could be employed to study the effect locally with a high temporal resolution and to observe temporal variations of the effect as well as its dependence on layer thickness and snow structure. Space-borne platforms, while limited by orbital mechanics and repeat-intervals, can provide a means to sample and to characterize the CBOE on the global scale. Finally, our characterization of the CBOE at X- and Ku-band in terrestrial snow could inspire future inter-planetary missions, aiming to search for water ice and possibly other types of snow, to employ bistatic radar measurements.

## 5 Conclusions

In this work we presented the first observations of the coherent backscatter opposition effect (CBOE) and the sampling of its angular peak shape at radio wavelengths within the Earth's cryosphere. The existence of the peak was confirmed in seasonal dry snow cover at Ku-band wavelengths by the ground-based bistatic radar system KAPRI. With the bistatic satellite formation TanDEM-X, the effect was also confirmed at X-band within the accumulation zone of high-altitude glaciers in the European Alps and the Karakorum.

The observability of the CBOE in bistatic radar measurements of snow presents an opportunity for future satellite missions aiming to derive snow properties from synthetic aperture radar data on the global scale. The radiometric precision requirement for such a spaceborne radar system is demanding, since the theoretical maximal amplitude of the effect is 3 dB – in this study, we were able to characterize the peak using TanDEM-X and data-driven radiometric calibration. Deployment of such bistatic systems – at bistatic angles up to one or two degrees, and covering the entire CBOE peak including the incoherent background – would open up a new pathway to characterize snow through microwave scattering.

The Ku-band observations presented in this paper suggest that the CBOE can be used as an indicator for presence of seasonal snow cover. At X-band, the CBOE could be applied to detect dry snow thicker than several meters, e.g. multi-year firn in accumulation areas of glaciers. Furthermore, through analysis of the angular width and height of the enhancement peak, scattering and absorption mean free paths within the snowpack can be estimated. Knowledge of the scattering mean free path and the discrimination between single (surface) and higher order (volume) scattering in areas where the CBOE is present could help to better constrain the radar penetration depth which, in turn, is crucial for precise surface height estimation by means of radar altimetry and interferometry.

The CBOE thus provides a pathway towards better characterization of area covered by snow, possibly also snow depth, and snow-water equivalent, which are the three key data products for snow as an Essential Climate Variable. Furthermore, the detection and characterization of the CBOE in terrestrial snow is also an encouraging sign for applying this measurement concept in space missions which aim to confirm the presence of water ice on surfaces of other solar system bodies.

*Data availability.* TanDEM-X data is available from DLR at https://tandemx-science.dlr.de/ and was provided by the proposal leinss_ XTI_ GLAC6870. KAPRI SLC data, as well as data values for plots, are published in the ETH Research Collection with the DOI 10.3929/ethz-b-000516171.

*Author contributions.* SL conceptualized the study of coherent backscatter enhancement in snow with bistatic radar. MS and SL designed the Davos field experiment and wrote the manuscript together. MS processed and analyzed the ground based data, SL the space borne data. IH suggested exploration of bistatic radar signals with KAPRI. All authors reviewed, edited, and approved the submitted version of the manuscript.

*Competing interests.* The authors declare that they have no conflict of interest.

*Acknowledgements.* The authors would like to thank Henning Löwe and one anonymous reviewer for their in-depth peer-reviews, and for constructive comments and suggestions how to improve the paper. Furthermore, the authors would like to thank Tingting Li, Yuta Izumi, Simone Jola, and Michael Arnold for their support during field work in Davos, Christian Mätzler and Rosemary Willatt for valuable feedback and discussion, Emmanuel Trouvé for proofreading the manuscript, Laurane Charrier for valuable input for applications of the CBOE, and Thomas Busche for support with the TanDEM-X acquisitions. The authors would also like to thank Mr. Reto Müller for access to his property during acquisitions and the staff of Bergbahnen Rinerhorn AG for assistance with equipment transportation. The TanDEM-X data was provided by the German Aerospace Center (DLR) via proposal XTI_GLAC6780. SL was supported by the French ministry of Europe and foreign affairs, within the framework of the 'Make Our Planet Great Again' program for postdoctoral researchers.

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
