# Peer review of "Coherent backscatter enhancement in bistatic Ku-/X-band radar observations of dry snow"

_The Cryosphere, 2021_

## Referee Comment (RC1)

**Review of "Coherent backscatter enhancement in bistatic Ku-/X-band radar observations of dry snow" by Stefko et al**

**Main comments**

The paper presents measurements of backscattering enhancement in snow by means of ground and space based Ku and X Band radar systems. The enhancement is a coherent "wave localization" effect through multiple scattering in a disordered microstructure. This effect is theoretically known for a long time, though it did not receive much attention in snow literature. To my knowledge, direct measurements of the enhancement peak in snow were never done before.

This is very creative and careful work and I recommend publication after a few minor things have been taken into account.

Kind regards,
Henning Löwe

**Minor comments**

(l73): This sounds as if CBOE may occur in pure water ice. I would mention the disorder here (e.g. porosity) too.

(l104): Its commonly termed *traditional* grain size.

(l115): missing spaces arount the hat symbol.

(l236): hat notation was already explained before.

(l298): Would be nice to state what's in fact the *meaning* of the porosity coefficient, besides giving its value.

(l.349): This information belongs rather into the method section.

(fig7): Maybe I missed it later in the discussion but what is the significance of the fact that $\Lambda_T$ estimate in summer is roughly the same as $\Lambda_T$ in winter? Since the backscattering enhancement is absent in summer, an interpretation of the summer data within this CBOE model should naively give an idea about the error of the parameters in winter. Why is the error on the length scale so drastically reduced?

(fig10): What does "supported by the horizontal gray line" mean in the caption? Is it a calculated line?

(sec 3.2.2): This is now a bit disappointing that no results are shown (in the main text) for the secondary observation site due to limited data. Why was the site chosen and introduced here at all? Maybe it should be dropped completely?

(l450): Was the corner reflector installed in summer?

(l445): "thickness of the snow layer" you mean snow depth?

(l448): "snowpacks thickness" $\rightarrow$ snow depth

(l494): Here, or before in sec 2.3, it would be helpful to discuss the relation of the effective transport parameters with the microstructure of the medium. What I grasp e.g. from Tsang, Ishimaru J. Opt. Soc. Am. A 1, 836 1984 is that the peak can be fully characterized by the effective (complex) wave propagation constant of the medium which can be computed from the two-point correlation function using common scattering approximations for snow. Is this sufficient or would a prediction of the profile (or prior estimates of the effective parameters) from in-situ measurements require more advanced structural information that is not yet measured in snow?

---

## Author Comment (AC1)

**Response to comments of Reviewer 1 (Henning Löwe) on manuscript tc-2021-358**

We would like to thank the reviewer for the in-depth peer-review, and for constructive comments and suggestions how to improve the paper. Below we give a point-by-point reply to individual comments. The reviewer's comments are labeled as **RC#** and the corresponding author responses are labeled **A#**. Suggestions to change or add in the manuscript are written in blue color.

**RC1:** (l73): This sounds as if CBOE may occur in pure water ice. I would mention the disorder here (e.g. porosity) too.

**A1:** We agree and propose to change the beginning of the paragraph at line 73 as follows:"In many of these experiments, observation of a backscatter enhancement peak at radio-frequencies was interpreted as the CBOE. This interpretation was then used to infer the possible existence of water ice (presumably with a porous or disordered structure so as to elicit the effect) on the surface of the corresponding solar system bodies."

**RC2:** (l104): Its commonly termed traditional grain size.

**A2:** We agree and propose to change the in line 104 the word "classical" to "traditional".

**RC3:** (l115): missing spaces around the hat symbol.

**A3:** Agree. We propose to add the spaces around the hat symbol.

**RC4:** (l236): hat notation was already explained before.

**A4** Agree. We propose to remove the sentence "The hat symbolˆindicates measured quantities."

**RC5:** (l298): Would be nice to state what's in fact the meaning of the porosity coefficient, besides giving its value.

**A5:** We agree. We propose to clarify the meaning of the porosity coefficient in section 2.3 as follows (see also author response **A13** for rationale and new proposed full text of section 2.3): "$K$ is a correction factor, described in (Hapke, 2012, p.164–167) as "porosity coefficent", which increases the extinction coefficient $E = S + A$ due to inter-particle effects in densely packed media where particles are large relative to $\lambda$. As ice grains are much smaller than the wavelength, we assume $K = 1$."

**RC6:** (l.349): This information belongs rather into the method section.

**A6:** We agree that it fits better into the method:data-selection section than into method:model-fitting section. We will move the sentence "From these, we removed 14 acquisitions for which TanDEM-X instead of TerraSAR-X acted as transmitter, resulting in slightly different antenna pattern that could not be 350 compensated through the calibration, especially in the HH polarization." to l.180 (method section 2.2.1) which will then read: "For 104 acquisitions TerraSAR-X acted as transmitter, for 14 acquisitions TanDEM-X acted as transmitter. We removed the 14 TanDEM-X acquisitions because they showed slightly different antenna patterns that could not be compensated through the calibration, especially at HH polarization, because of a too small number of acquisitions. For the remaining 104 acquisitions, bistatic baselines ..."

**RC7:** (fig7): Maybe I missed it later in the discussion but what is the significance of the fact that $\Lambda_T$ estimate in summer is roughly the same as $\Lambda_T$ in winter? Since the backscattering enhancement is absent in summer, an interpretation of the summer data within this CBOE model should naively give an idea about the error of the parameters in winter. Why is the error on the length scale so drastically reduced?

**A7:** We interpret the fact that the best estimates of $\Lambda_T$ are roughly the same for summer and winter ($\sim 0.4\,\mathrm{m}$) as a coincidence, that can not be assigned a strong physical interpretation for two

reasons. Firstly, the estimate of $\Lambda_T$ for summer has a much larger confidence interval which reduces the significance of the numerical match of the best estimate. Furthermore, due to the non-linearity of the model, the behaviour of the errors of one parameter is affected by the current value of the second parameter. As a specific example of the summer scenario, when the value of the estimate of the absorption mean free path $\Lambda_A$ is low (0.4 m, i.e. comparable to or lower than the scattering mean free path $\Lambda_T$), the model becomes much less sensitive to variations of the value of $\Lambda_T$. Since $\Lambda_T$ can be interpreted as a measure of the relative width of the intensity peak, its precise value becomes non-physical when no enhancement peak is present. This also makes drawing conclusions from direct comparison of the errors of $\Lambda_T$ estimates between summer and winter difficult, because the $\Lambda_A$ value is diametrically different in the two cases. To clarify this, we propose the following changes:

- Line 361-365: Reorder the paragraph to address the winter dataset first, and provide more information about the summer dataset model fit: "For the winter dataset a clear intensity peak is detected, with a HWHM of approx. 0.25° and amplitude $B_C(0) \approx 0.5$ (1.8 dB), corresponding to $\Lambda_T \approx (0.4 \pm 0.1)$ m for the HH and VV polarization. The derived absorption lengths $\Lambda_A$ are much longer than the scattering lengths with $\Lambda_A \approx (11 \pm 7)$ m for the HH polarization and $\Lambda_A \approx (19 \pm 12)$ m for the VV polarization. For the summer dataset, the flat profile of the observed intensity curve indicates that very little or no backscatter enhancement is present – this is reflected in the model fit in the low value and large confidence interval (relative to the value) of the absorption length $\Lambda_A \approx (0.4 \pm 0.4)$ m. The estimates of the scattering length in the summer dataset ($\Lambda_T \approx (0.40 \pm 0.27)$ m and $\Lambda_T \approx (0.43 \pm 0.39)$ m for the VV and HH polarization respectively) have comparable value to the winter estimates, however the much larger confidence intervals indicate that the value of $\Lambda_T$ could not be determined more precisely for the summer dataset due to the absence of a clear enhancement peak. The uncertainty of the value estimates corresponds to the 95% confidence interval."

- Fig. 7: After the first sentence, add the following sentence: "For the summer dataset (orange), the comparable values of the $\Lambda_T$ and $\Lambda_A$ estimates, as well as their relatively large confidence intervals, indicate that the CBOE peak was not detectable."

- In the Discussion section 4.2.1, modify the first paragraph by replacing the lines 434-437 (Starting at "The absence..." until the end of the paragraph) with the following text: "In the summer scenario (i.e. absence of a clear backscatter enhancement peak), the model, described in Sect. 2.3 and visualized in Fig. 6, predicts that the absorption length is shorter or equal to the scattering length ($\Lambda_A \leq \Lambda_T$). An interpretation of this scenario is that higher-order scattering paths are suppressed due to absorption, and thus the summer scenario is dominated by a single-scattering process. In the summer scenario the model becomes much less sensitive to the precise value of $\Lambda_T$ (which is a measure of the width of the peak), and thus estimates of this value have much higher uncertainty as opposed to the case of a clearly detectable enhancement peak in winter."

**RC8:** (fig10): What does "supported by the horizontal gray line" mean in the caption? Is it a calculated line?

**A8:** In an earlier version of the processing, we used a manually chosen threshold on the backscatter intensity at $-8$ dB to remove data with wet snow. However, this threshold was replaced by the temporal constraint 01 Dec - 31 May. Therefore we propose to remove the gray line from Fig. 10 and to remove the statement "(supported by the horizontal gray line)" from the figure caption.

**RC9:** (sec 3.2.2): This is now a bit disappointing that no results are shown (in the main text) for the secondary observation site due to limited data. Why was the site chosen and introduced here at all? Maybe it should be dropped completely?

**A9:** The secondary site was chosen in hope to see a stronger effect because of possibly larger bistatic angles. However, the bistatic angles were very similar here. Nevertheless, the analysis confirmed that CBOE also occurs in the accumulation zone of Teram-Shehr/Rimo glacier. The observed enhancement was slightly stronger, but no additional information was gained from the secondary site. We propose, therefore, to move almost all information regarding the secondary test site from the main part of the manuscript to the supplements. As the secondary site still confirms our observations we will keep a reference at line 175 "As a best compromise, we selected the Jungfrau-Aletsch region in Switzerland but also analyzed the Teram-Shehr/Rimo glacier in the Karakorum (supplementary material) where a considerably lower number of acquisitions is available.". In addition, we will add in line 462 "These observations were confirmed by the data from the Teram-Shehr/Rimo glacier in the Karakorum (see supplements)." and remove "as well as for Teram-Shehr/Rimo glacier the Karakorum (Figs. S10 and S11)." above (l.460).

**RC10:** (l450): Was the corner reflector installed in summer?

**A10:** We suggest to change l.450 "a corner reflector at the bottom of a 1.55m deep snow pit" to "a corner reflector lowered to the bottom of a 1.55m deep snow pit with vertical walls" to clarify that the reflector was installed in winter.

**RC11:** (l445): "thickness of the snow layer" you mean snow depth?

**A11:** We propose to change the wording in following lines:

- Line 445: "thickness of the snow layer" → "snow depth"

- Line 446: "the snow layer thickness" → "snow depth"

- Line 448: "ground layer" → "ground"

**RC12:** (l448): "snowpacks thickness" → snow depth

**A12:** We agree and propose to simplify the sentence to: "Nevertheless, the snow depth of only 3–4 scattering mean free paths could limit higher order scattering."

**RC13:** (l494): Here, or before in sec 2.3, it would be helpful to discuss the relation of the effective transport parameters with the microstructure of the medium. What I grasp e.g. from Tsang, Ishimaru J. Opt. Soc. Am. A 1, 836 1984 is that the peak can be fully characterized by the effective (complex) wave propagation constant of the medium which can be computed from the two-point correlation function using common scattering approximations for snow. Is this sufficient or would a prediction of the profile (or prior estimates of the effective parameters) from in-situ measurements require more advanced structural information that is not yet measured in snow?

**A13:** Yes, we agree and will describe the relation between the effective transport parameters $\Lambda_A, \Lambda_T$ and the microstructure of snow using references to empirical data (Wiesmann et al., 1998; Wiesmann and Mätzler, 1999) and also to the modelling work. To keep the model-section concise, we add a brief reference to the SMRT model (Picard et al., 2018) and will add a new discussion section (4.4) to provide further details. Providing this relation requires putting the model from (Hapke, 2012; Akkermans et al., 1986) into context with earlier models (e.g. (Tsang and Ishimaru, 1984, 1985)) and outlining assumptions of current models for CBOE. Based on the current state-of-the art of microwave backscatter models for snow (e.g. (Picard et al., 2018)) we can then provide a link of the transport parameters to the snow microstructure.

Regarding prediction of the peak profile from in-situ measurements: We would like to point out that current CBOE models assume mostly a half-space filled with a medium with homogeneous scattering/absorption properties which is not the case for a natural, layered snow pack where scattering can also occur at the interfaces between different snow layers and also at density-fluctuations in

the horizontal direction (Proksch et al., 2015). Even for homogeneous snow slabs, results from the characterization of the microstructure with different methods, as well as the application of different microwave scattering models, can differ significantly (Vargel et al., 2020). To avoid overcomplication of the model we consider the multi-layer snow pack as a homogeneous semi-infinite scattering medium but would like to point out, that the derived parameters $\Lambda_T, \Lambda_A$ might require additional correction factors when a scattering model for CBOE in snow at microwave frequencies becomes available or when the limited thickness of the snow pack becomes relevant, i.e. when its optical thickness becomes relevant (see comment from Reviewer 2).

In order to provide the above discussed information in the manuscript, we suggest to rework the model-section (2.3), to add to the discussion a subsection "4.4 Link to the microstructure of snow", and to add a new discussion subsection "4.5 Limitations of the model". The complete proposed new text of these sections is shown in the itemized list below:

[revised manuscript text omitted]

We also propose the following changes to further address this reviewer comment:

- Based on the model limitations, we like to adjust the sentence in line 494 to: "When prior estimates (...) are available, (...) to roughly estimate (...) where the CBOE might affect the measurements (see also Sect. 4.4)."

- To provide a more extensive historical context, we also propose to add (Tsang and Ishimaru, 1984) into the reference list at line 32.

- We propose to add sentences pointing out the good match between our observations and observations by Wiesmann et al. (1998) into appropriate places in sections 4.2.1 and 4.2.2

**References**

[revised manuscript text omitted]

---

## Author Comment (AC2)

**Response to comments of Reviewer 2 on manuscript tc-2021-358**

We would like to thank Reviewer 2 for the peer-review, and for constructive comments and suggestions how to improve the paper. Furthermore, we thank Reviewer 2 for pointing to additional references for early experimental work regarding backscatter enhancement. Below we give a point-by-point reply to individual comments. The reviewer's comments are labeled as **RC#** and the corresponding author responses are labeled **A#**. Suggestions to change or add in the manuscript are written in blue color.

**RC1:** For Ku band volume scattering of snow, cross polarizations are usually strong. In laboratory experiments cross polarization enhancement are more conspicuous than co-polarization for both volume scattering (Kuga et al. JOSA A, 1985) and surface scattering ( Johnson et al. IEEE Transactions on Antennas and Propagation 1994). What are the reasons for non-observations in this paper. How about deeper snow?

**A1:** In the reference Kuga et al. (1985), experimental data show (Figs. 2-4) that the co-polarized (CP) enhancement peak is more pronounced than the cross-polarized (XP) peak. Fig. 5, which shows a more pronounced XP peak, is a plot of a theoretical model, which doesn't seem to be experimentally validated. The authors themselves state that "It should be noted that Fig. 5 is obtained using the Rayleigh point-dipole model and cannot be compared quantitatively with the experimental results in which finite-sized particles are used." Furthermore, later theoretical references regarding CBOE (Mishchenko, 1992a, Figs. 3,4,12,13) as well as experiments (Wolf and Maret, 1985, Fig. 4) show that the co-polarized enhancement factor is larger than the cross-polarized enhancement factor (except extreme incidence angles very close to 90deg), and that with thicker samples, the co-polarized enhancements increases while the cross-polarized enhancment decreases and van Albada et al. (1987). The interpretation of the CBOE being caused by constructive interference of time-reversed pairs of electromagnetic wave paths also supports stronger intensities in the co-polarized channels, since if the incoming and returning waves have orthogonal polarizations, the time-reversal symmetry of the two opposing-direction paths is broken and constructive interference doesn't necessarily occur. We thus expected (and confirmed in preliminary analysis) that the cross-polarized enhancement peak is much smaller than the clearly pronounced co-polarized peak, despite volume scattering being the cause of its presence.

The enhancement peaks of rough surfaces experimentally observed by Johnson et al. (1996) in both co-polarized and cross-polarized channels appear to have half-width-at-half-maximum (HWHM) of at least 5–10 degrees, which is much larger than the characteristic peak width of the CBOE. Furthermore, the CBOE is not characteristic for surface scattering processes where low-order of scattering is much more likely. We thus postulate that the enhancement peaks observed by Johnson et al. (1996) are not caused by the CBOE, and thus no conclusions about the cross-polarized behaviour of the CBOE enhancement peaks observed in our experiment can be made from the referenced publication.

The reason for not including cross-pol KAPRI data in the results is the combination of the enhancement peak being less pronounced in the cross pol channels (as explained above), and the cross-pol data being much closer to the noise floor of the experiment. While it is true that – compared to other more surfaces-like media – snow is considered a medium where volume scattering is considerable, the cross-polarized channels still exhibit lower overall backscatter intensities than the co-polarized channels – e.g. in the experiments of King et al. (2015) the measured co-polarized $\sigma_0$ of snow at Ku-band varied between $-13$ and $-5$ dB, while the cross-polarized value varied between $-26$ and $-15$ dB. Furthermore, while the monostatic version of KAPRI offers in general very good SNR, the SNR of the bistatic experimental setup is reduced due to the necessity of using lower-gain horn antennas for reception. For the winter experiment, the SNR from the ROI for co-polarized channels for a single acquisition was estimated from the data as approximately 15 dB. For the cross-polarized channels, this was reduced to approximately 9 dB. Combined with the lower amplitude of the enhancement peak,

and other sources of inaccuracy mentioned in Sect. 2.1.3, this was deemed too low to allow precise analysis of the existence of the peak and its properties, as noted on line 117.

Regarding deeper snow: Increasing the snow depth in our setting of 2 m of seasonal snow will likely exhibit diminishing effects on the observed cross- and co-pol backscatter because our retrieved scattering mean free path value of ~40 cm suggests that most of the scattering occurs in the uppermost ~2 m of the snow layer.

Future experiments aimed at acquisitions with increased SNR can indeed provide further insights based on analysis of cross-polarized channels, as we note in Section 4.4.

To consider the reviewer's comment, we suggest to add to the model section the following text: "Most CBOE models are based on scalar waves which do not consider the vector character of electromagnetic waves, i.e. their polarization. However, experimental and theoretical works show that CBOE occurs predominantly for co-polarized transmitted and received waves (VV and HH) where the model matches well experimental observation. They also show that CBOE for cross-polarized (VH) observations is significantly weaker and decreases with increasing sample thickness (van Albada et al., 1987; Mishchenko, 1992a,b; Wolf and Maret, 1985)."

To reiterate the point in the discussion we also suggest to add these references to line 526 which will then read: "In terms of polarimetric measurements, the results of this study, as well as experimental work and theoretical models (van Albada et al., 1987; Mishchenko, 1992a,b; Wolf and Maret, 1985), indicate that the effect is present predominantly in co-polarized channels, and the effect is equally strong at both horizontal and vertical polarizations."

To explain why we do not snow cross-pol observations for TanDEM-X, we suggest to add to the description of the Aletsch dataset in section 2.2.1: "At VH polarization no acquisitions at sufficiently large $\beta$ were available."

**RC2:** The optical thickness can indicate the order of multiple scattering. Please discuss the optical thicknesses tau in the measurements at X band and Ku band.

**A2:** This is already discussed in the last paragraph of section 4.2.1 for Ku-band and in the second and third paragraph of section 4.2.2 even though the term "optical thickness" is not explicitly mentioned. In addition to (Van Der Mark et al., 1988) and (Van Albada et al., 1988) that are already referenced in our paper, we suggest to reference the work of Tsang and Ishimaru (1985) who modeled that the enhancement is decreased when the optical thickness $\tau = E\,d < 4$ with extinction coefficient $E$ and sample thickness $d$. To address this, we suggest to add to section 4.2.1:

"Nevertheless, the optical thickness $\tau_d = E\,d \approx d/\Lambda_T$ of the snow depth $d$ of only 3–4 scattering mean free paths $\Lambda_T$ could limit higher order scattering. While Tsang and Ishimaru (1985) conclude that already at $\tau_d = 4$ models approximate well the half-space solution (where $\tau_d = \infty$), Van Der Mark et al. (1988, Figs. 9,12) show that the peak height and width, at least for very weakly absorbing media ($\Lambda_A \gg \Lambda_T$), might be affected up to $\tau_d \approx 30$."

and to write more clearly in Sect. 4.2.2: "On the tongue of Great Aletsch Glacier, where a seasonal snowpack is present during winter, no backscatter enhancement was observed in X-band (Fig. 9c). As seasonal snow is younger than multi-year firn, smaller snow grain sizes are expected, resulting in scattering lengths larger than the value $\Lambda_T = 2.1$ m determined for the accumulation area. The thickness of the seasonal snowpack of 0–3 m corresponds therefore to an optical thickness $\tau_d \approx 1$ or less, which considerably affects the peak intensity (Van Der Mark et al., 1988, Fig. 9). In consequence, the single scattering at the (possibly rough) snow-ice interface at the bottom of the snowpack can remain the dominant scattering process. The low average number of scattering events in the seasonal snow volume is, therefore, not sufficient for the CBOE to occur on the ablation area of Great Aletsch Glacier. "

In the new section about limitations of the model (see comments to Reviewer 1), we suggest to address the limited optical thickness of the snow pack again and will write: "An additional limitation for a accurate estimation of $\Lambda_A$, possibly also $\Lambda_T$, results from the assumption that the scattering

medium fills a semi-infinite space whereas the snow pack has a limited optical thickness $\tau_d$. Hence, $\Lambda_A$ might be underestimated due to limited layer thickness (Van Der Mark et al., 1988; Van Albada et al., 1988)."

**RC3:** For X band at Tandem X, the soil surface below the snow have significant contributions. What is the magnitude of surface scattering of the snow/soil interface below the snow layer?

**A3:** Comparing the scattering mean free path at X-band (here: 1-3 meters) with the typical snow height in the Swiss Alps in winter ($\sim 1 - 4\,\mathrm{m}$) indicates that there must be a strong contribution (likely more than 50%) from the ground, at least for snow over ground or snow over ice. This is discussed already in section 4.2.2 (with a modification proposed in **A2**): "In consequence, the single scattering at the (possibly rough) snow-ice interface at the bottom of the snowpack can remain the dominant scattering process."

**RC4** Will there be coherent backscattering due to rough soil surface below the snow at X band?

**A4:** As indicated in Fig. 9 (and also Fig. 11), and as discussed in section 4.2.2 (Satellite observations - TanDEM-X) we did not observe any coherent backscatter from areas different than the high accumulation area of glaciers. Specifically, Figure 9 shows the dependence of the bistatic-to-monostatic backscatter ratio for different areas (high accumulation area, glacier ablation zone, conifer forest). Figure 9c for the glacier ablation zone, where 2-4 meters of snow cover the rough ice surface, does not show any signal of coherent backscatter enhancement. To make this explicitly clear, we suggest to add "We also did not observe coherent backscatter enhancement in any area other than the high accumulation area, even though the tongue of Aletsch glacier is highly crevassed and valley slopes are covered by rock debris. From this we conclude that in the X-band, rough surfaces do not elicit the CBOE." at the end of section 4.2.2.

**References**

Johnson, J. T., Tsang, L., Shin, R. T., Pak, K., Chan, C. H., Ishimaru, A., and Kuga, Y.: Backscattering enhancement of electromagnetic waves from two-dimensional perfectly conducting random rough surfaces: A comparison of monte carlo simulations with experimental data, IEEE Transactions on Antennas and Propagation, 44, 748–756, https://doi.org/10.1109/8.496261, 1996.

King, J., Kelly, R., Kasurak, A., Duguay, C., Gunn, G., Rutter, N., Watts, T., and Derksen, C.: Spatio-temporal influence of tundra snow properties on Ku-band (17.2 GHz) backscatter, Journal of Glaciology, 61, 267–279, https://doi.org/10.3189/2015JoG14J020, 2015.

Kuga, Y., Tsang, L., and Ishimaru, A.: Depolarization effects of the enhanced retroreflectance from a dense distribution of spherical particles, Journal of the Optical Society of America A, 2, 616, https://doi.org/10.1364/JOSAA.2.000616, 1985.

Mishchenko, M. I.: Polarization characteristics of the coherent backscatter opposition effect, Earth, Moon and Planets, 58, 127–144, https://doi.org/10.1007/BF00054650, 1992a.

Mishchenko, M. I.: Enhanced backscattering of polarized light from discrete random media: calculations in exactly the backscattering direction, J. Opt. Soc. Am. A, 9, 978–982, https://doi.org/10.1364/JOSAA.9.000978, 1992b.

Tsang, L. and Ishimaru, A.: Theory of backscattering enhancement of random discrete isotropic scatterers based on the summation of all ladder and cyclical terms, Journal of the Optical Society of America A, 2, 1331, https://doi.org/10.1364/JOSAA.2.001331, 1985.

van Albada, M. P., van der Mark, M. B., and Lagendijk, A.: Observation of weak localization of light in a finite slab: Anisotropy effects and light path classification, Phys. Rev. Lett., 58, 361–364, https://doi.org/10.1103/PhysRevLett.58.361, 1987.

Van Albada, M. P., Van Der Mark, M. B., and Lagendijk, A.: Polarisation effects in weak localisation of light, Journal of Physics D: Applied Physics, 21, S28–S31, https://doi.org/10.1088/0022-3727/21/10S/009, 1988.

Van Der Mark, M. B., van Albada, M. P., and Lagendijk, A.: Light scattering in strongly scattering media: multiple scattering and weak localization, Physical Review B, 37, 3575, https://doi.org/10.1103/PhysRevB.37.3575, 1988.

Wolf, P. E. and Maret, G.: Weak localization and coherent backscattering of photons in disordered media, Physical Review Letters, 55, 2696–2699, https://doi.org/10.1103/PhysRevLett.55.2696, 1985.

---

## Author Response (AR1)

**Author response for manuscript tc-2021-358**

We thank the reviewers for their in-depth reviews of the manuscript. Responses to their reviews and the proposed changes based on their suggestions are listed in detail in Author Comments:

- https://doi.org/10.5194/tc-2021-358-AC1

- https://doi.org/10.5194/tc-2021-358-AC2

Furthermore, we would like to propose the following changes (additions highlighted in blue):

- Add a phrase to the abstract: "In the VV polarization, we obtained ... at X-band, assuming an optically thick medium". This is in order to reflect the expanded snow modeling considerations based on inputs from Reviewer 1.

- In section 2.3.3, we propose to expand the explanation of the likely cause of temporal deviations: "The deviations might originate from bright azimuth or range-ambiguities or areas affected by sidelobes of layover" → "The deviations might partially originate from bright azimuth ambiguities. We also believe that double-reflections occurring within layover, with a reflection on each side of a north-south oriented valley and an additional propagation path between the two reflections, cause further radar echos appearing beyond the layover area. These artifacts appear stronger at HH than at VV (due to reflections close to the Brewster-angle) and are also stronger with wet snow due to more specular reflection compared to dry snow or summer with more diffuse reflections..."

- In section 2.2.1, we propose to add at the end of the second paragraph a note describing our recent measurements from March 2022 in the region of interest: "At the beginning of March 2022, we measured snow temperatures of $-12 \pm 3\,°\mathrm{C}$ in the upper two meters and $-4 \pm 1\,°\mathrm{C}$ at -5 m at 3640 m altitude (46.5515°N, 8.0062°E). Both Bannwart's firn cores as well as our snow pit measurements indicate the presence of a few cm thick ice layer resulting from melt and refreeze during previous summers below the several meter thick seasonal snow cover."

The submitted revised version incorporates all these changes, as well as minor edits to improve the readability and clarity of the manuscript.